# The MuvB complex binds and stabilizes nucleosomes downstream of the transcription start site of cell-cycle dependent genes

Anushweta Asthana [1], Parameshwaran Ramanan[1], Alexander Hirschi[1], Keelan Z. Guiley[1], Tilini U. Wijeratne[1], Robert Shelansky[2], Michael J. Doody[2], Haritha Narasimhan[1], Hinrich Boeger[2], Sarvind Tripathi [1], Gerd A. Müller [1✉] & Seth M. Rubin [1✉]

The chromatin architecture in promoters is thought to regulate gene expression, but it remains uncertain how most transcription factors (TFs) impact nucleosome position. The MuvB TF complex regulates cell-cycle dependent gene-expression and is critical for differentiation and proliferation during development and cancer. MuvB can both positively and negatively regulate expression, but the structure of MuvB and its biochemical function are poorly understood. Here we determine the overall architecture of MuvB assembly and the crystal structure of a subcomplex critical for MuvB function in gene repression. We find that the MuvB subunits LIN9 and LIN37 function as scaffolding proteins that arrange the other subunits LIN52, LIN54 and RBAP48 for TF, DNA, and histone binding, respectively. Biochemical and structural data demonstrate that MuvB binds nucleosomes through an interface that is distinct from LIN54-DNA consensus site recognition and that MuvB increases nucleosome occupancy in a reconstituted promoter. We find in arrested cells that MuvB primarily associates with a tightly positioned +1 nucleosome near the transcription start site (TSS) of MuvB-regulated genes. These results support a model that MuvB binds and stabilizes nucleosomes just downstream of the TSS on its target promoters to repress gene expression.

[1] Department of Chemistry and Biochemistry, University of California, Santa Cruz, CA 95064, USA. [2] Department of Molecular, Cell, and Developmental Biology, University of California, Santa Cruz, CA 95064, USA. ✉email: gemuelle@ucsc.edu; srubin@ucsc.edu

Chromatin architecture and the position of nucleosomes influence DNA-mediated processes including the transcription of genes[1]. Transcription by RNA polymerase results in significant changes to nucleosome positioning, as the basal transcription machinery must overcome the energetic barriers presented by the placement of nucleosomes along promoters and the gene body[2–4]. RNA polymerase with the aid of elongation factors and histone chaperones can bind and evict octamer proteins or reposition nucleosomes present in the gene body to access the gene for transcription. In addition, chromatin remodelers and histone-modifying enzymes are thought to facilitate or inhibit transcription by arranging or displacing nucleosomes near transcription start sites, by altering the packing of nucleosomes, and by modulating the affinity of histone proteins for the DNA backbone. Less is known about how nucleosomal architecture is influenced by the activity of transcription factors (TFs). While recent evidence shows that pioneer TFs can bind target DNA sites within the nucleosome wrap and recruit remodelers to alter chromatin architecture, other TFs compete with nucleosomes for access to their DNA consensus sequence[5–7]. A thorough molecular description of how many regulatory TFs cooperate and engage with nucleosomes to modulate gene-expression remains elusive.

The MuvB TF complex binds to target gene promoters and regulates a large set of cell-cycle genes. MuvB temporally coordinates the expression of genes necessary for DNA synthesis, centromere construction, mitotic division, and cell-cycle exit[8–10]. In mammals, cell-cycle-dependent gene expression occurs primarily in two waves of transcription, which take place around the G1/S and G2/M transitions and depend on the activity of MuvB and the other TFs E2F, B-MYB, and FOXM1[11–15]. These TFs and their regulators are commonly deregulated in cancer[16–18].

The MuvB complex, components of which are evolutionarily conserved throughout animals and ciliates, plays a key role in development and differentiation and is an essential regulator of cell-cycle-dependent gene expression programs[9,10,19–23]. During quiescence and in early G1, MuvB binds to the retinoblastoma protein (RB) paralogs p130 or p107 (p130/p107) and E2F4-DP. This complex, known as DREAM, represses S phase genes and late cell-cycle genes[22,24,25]. Upon entry into the cell cycle, cyclin-dependent kinases along with their cyclin partners phosphorylate and release p130/p107 from the MuvB core, disassembling DREAM but keeping the core MuvB intact[22,25–27]. During S phase, the MuvB core binds to the proto-oncoprotein B-MYB and forms the MYB-MuvB (MMB) complex, which in concert with FOXM1 functions as a transcriptional activator of G2/M genes[22,23,28]. While the cellular imbalance of activating and repressive MuvB complexes is associated with several cancers[29,30], the molecular details of MuvB assembly and function are poorly understood.

The core MuvB complex is composed of the five proteins LIN9, LIN37, LIN52, LIN54, and RBAP48 (or RBBP4). MuvB is localized to its target cell-cycle genes through LIN54, which binds target promoters directly at a consensus DNA sequence[31–33]. The short sequence motif, known as the cell-cycle genes homology region (CHR), is found in close proximity to the transcription start site (TSS) and is often located just downstream of a truncated E2F binding site, known as the cell-cycle-dependent element (CDE)[34]. LIN52 is a transcription factor adapter protein that recruits either B-MYB or p130/p107, depending on cell-cycle phase[35,36]. RBAP48 is a histone binding chaperone protein that is found in several complexes that interact with chromatin, including CAF-1, NuRD, PRC2, and SIN3-HDAC[37–40]. In mammals, RBAP48 has a highly similar (89% sequence identity) paralog named RBAP46 (or RBBP7), which has not been identified in complexes with MuvB components[22]. Both proteins are

found in chromatin remodeler complexes, sometimes together. Less is known regarding the structure and biochemical function of LIN9 and LIN37, although a LIN9 sequence near its C-terminus co-folds with LIN52 to create the B-MYB-binding site[35].

Genetic evidence suggests that MuvB core proteins are essential in regulating cell-cycle-dependent gene expression. In flies and worms, knockout of MuvB components contributes to inappropriate derepression of developmental gene programs[19–21]. In mammals, LIN9 is essential for the expression of G2/M genes; loss of LIN9 causes mitotic defects and is embryonically lethal in mice[41,42]. On the other hand, knockdown of LIN9 in cell culture results in compromised repression of DREAM target genes upon induced cell-cycle exit[22]. Knockout of the MuvB subunit LIN37 results in loss of MuvB-mediated gene repression in G0 and G1, but it does not lead to any observable changes in MYB-MuvB (MMB) mediated gene expression in G2/M[24]. Similarly, RNAi depletion of the *Drosophila* ortholog of RBAP48 specifically results in a derepression of dE2F2 target genes but does not result in defects in proliferation or gene expression[43]. These findings implicate MuvB core subunits in both positively and negatively modulating gene-expression, yet the biochemical mechanism behind their function remains unknown.

Here we investigated how MuvB represses gene expression, with emphasis on characterizing the structure and function of LIN9, LIN37, and RBAP48. We demonstrate that LIN9 and LIN37 together form an essential scaffold that holds together the core complex, and we determined a crystal structure that reveals how they together recruit RBAP48. We show that through RBAP48, MuvB binds directly to nucleosomes, either by interacting with H3 tails or the core particle. Using single-molecule electron microscopy, we found that MuvB increases nucleosome occupancy in a reconstituted cell-cycle gene promoter. These data indicate that MuvB associates with and stabilizes nucleosomes in the absence of other factors. Finally, we implemented a protocol that applies micrococcal nuclease digestion of chromatin and co-precipitation (MNase-ChIP) to study interactions of MuvB with nucleosomes in HCT116 cells. Our results support a model that MuvB binds to nucleosomes near the transcription start sites of target genes and stabilizes nucleosomes to repress cell-cycle-dependent gene expression.

## Results

**LIN9 and LIN37 are together required for assembly of MuvB.** Beyond the role of LIN9 in binding B-MYB, the structure and biochemical function of LIN9 and LIN37 have not been previously characterized. Human LIN37 is a 246 amino acid protein that has no homology to any known structures. Sequence analysis suggests the presence of several short, structured regions (1–43, 95–126, 203–246) that are interspersed with sequences that are likely disordered (Fig. 1a). The segment 95–126, which we call the CRAW domain for the presence of a CRAW amino acid sequence, is highly conserved among animal orthologs and is necessary for LIN37 assembly into MuvB and for its activity in gene repression[24]. Human LIN9 contains 542 amino acids, and beyond the presence of a Tudor domain, it also exhibits no homology to known structures (Fig. 1a). The N-terminal ~90 amino acids of LIN9 are poorly conserved and have no predicted structure. The segment from 94 to 278 (previously called the domain in RB-related pathway or DIRP; Pfam 06584) contains the Tudor domain and is conserved between MuvB and the related tMAC complex[44]. The helical segment between 333 and 421 forms the MYB-binding domain (MBD) together with LIN52[35], while the C-terminus (residues 428–542) also has predicted helical structure.

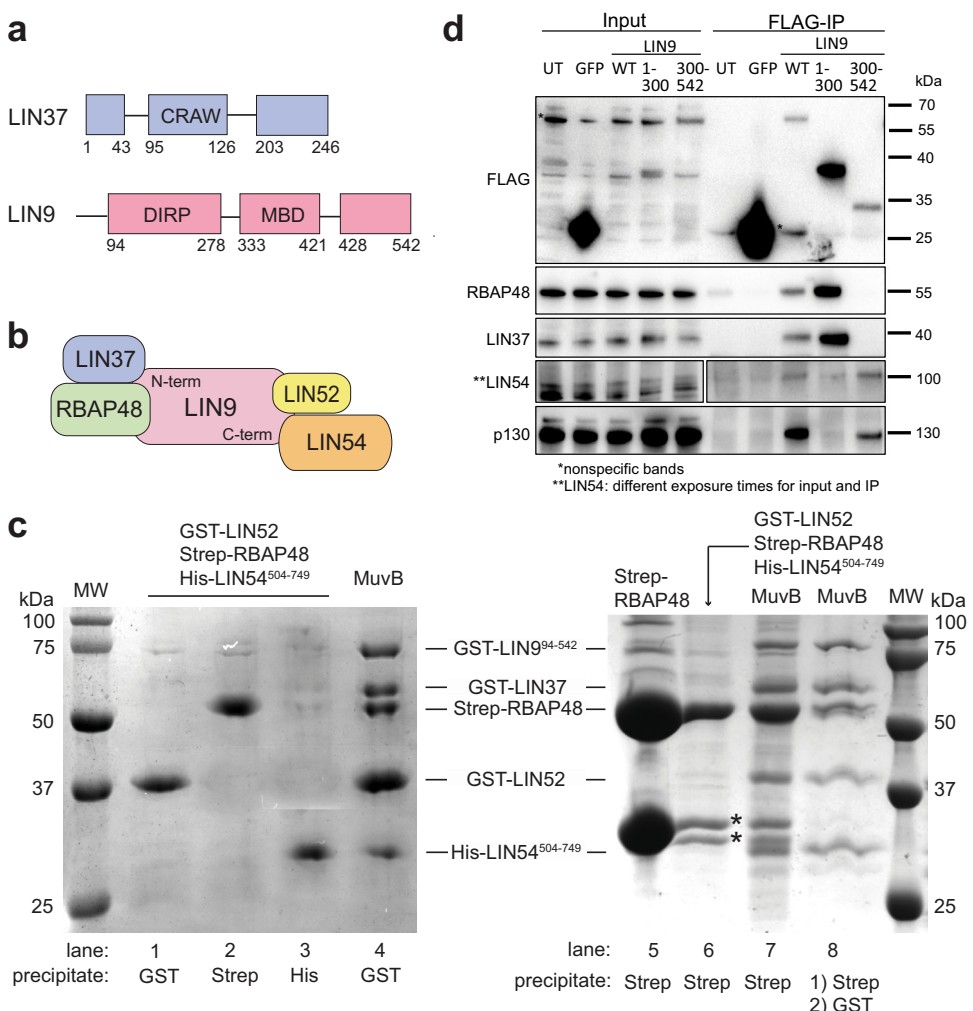

**Fig. 1 LIN9 and LIN37 scaffold the MuvB complex. a** Domain architecture of human LIN9 and LIN37, with regions of predicted and validated structure shown as blocks. The conserved LIN37 CRAW domain and LIN9 DIRP domain structures are determined here. MBD is the MYB-binding domain. **b** Schematic model for subunit interactions within MuvB. **c** The indicated tagged subunits or MuvB complex (GST-LIN52, Strep-RBAP48, His-LIN54[504-749], GST-LIN37, and GST-LIN9[94-542]) were expressed in Sf9 cells and extracts were precipitated with resin capturing the indicated tag. Proteins were visualized with coomassie staining. *Indicates impurities or degradation observed in some RBAP48 expressions. These bands are not pulled out from the tandem purification. The experiment was repeated three times with similar results. **d** HCT116 cells were transfected with plasmids encoding the indicated FLAG-tagged mouse protein. FLAG-tagged proteins were precipitated from extracts using anti-FLAG antibody and visualized with anti-FLAG immunoblotting and immunoblotting with antibodies that recognize RBAP48, LIN37, LIN54, and p130. A biological replicate of the experiment was performed, and results were similar.

Considering previous observations that LIN9 binds directly to multiple core MuvB and MuvB-interacting proteins[23,27,35] and that LIN9 knockdown results in DREAM complex assembly defects in T98G cells[22], we hypothesized that LIN9 is a scaffold onto which the other proteins assemble (Fig. 1b). To probe MuvB complex assembly in a reconstituted system, we performed co-precipitation experiments by expressing human proteins with different affinity tags in Sf9 insect cells (Fig. 1c). We expressed full-length RBAP48, LIN52, and LIN37 and the relatively conserved and structured regions of LIN9 (residues 94–542, called LIN9[94-542]) and LIN54 (residues 504–749, LIN54[504-749]). When the three MuvB components RBAP48, LIN52, and LIN54 were co-expressed, we did not see co-precipitation (Fig. 1c, lanes 1–3, 6). In contrast, we were able to reconstitute the MuvB complex when all five components were co-expressed (Fig. 1c, lanes 4 and 7), and we could demonstrate co-elution as a single complex by performing successive precipitations of different affinity tags (Fig. 1c, lane 8). In our baculovirus system, we were unable to express LIN9 in the absence of LIN37, so we could not

test whether LIN9 alone is required in our reconstitution. However, it has previously been reported that DREAM and MuvB complexes are able to assemble in the absence of LIN37[19,24,45]. Taken together, these results suggest that the LIN9 subunit of MuvB coordinates RBAP48, LIN52, and LIN54 to assemble the complex.

To further probe how LIN9 interactions with the other MuvB subunits organize the overall architecture of the complex, we expressed Flag-tagged mouse LIN9 constructs in HCT116 cells and analyzed binding by co-immunoprecipitation (Fig. 1d). We observed differences in the interactions made by LIN9[1-300], which contains the DIRP domain, and the interactions made by LIN9[300-542], which contains the MYB-binding domain and C-terminus. Only LIN9[300-542] co-precipitated p130. This observation is consistent with the known direct association of LIN9[MBD] with LIN52 and the direct association of the LIN52 N-terminus with p130[26,35]. LIN9[300-542] also associates with LIN54, whereas LIN9[1-300] does not immunoprecipitate LIN54 above background in our experiment. In contrast, only LIN9[1-300] co-precipitated

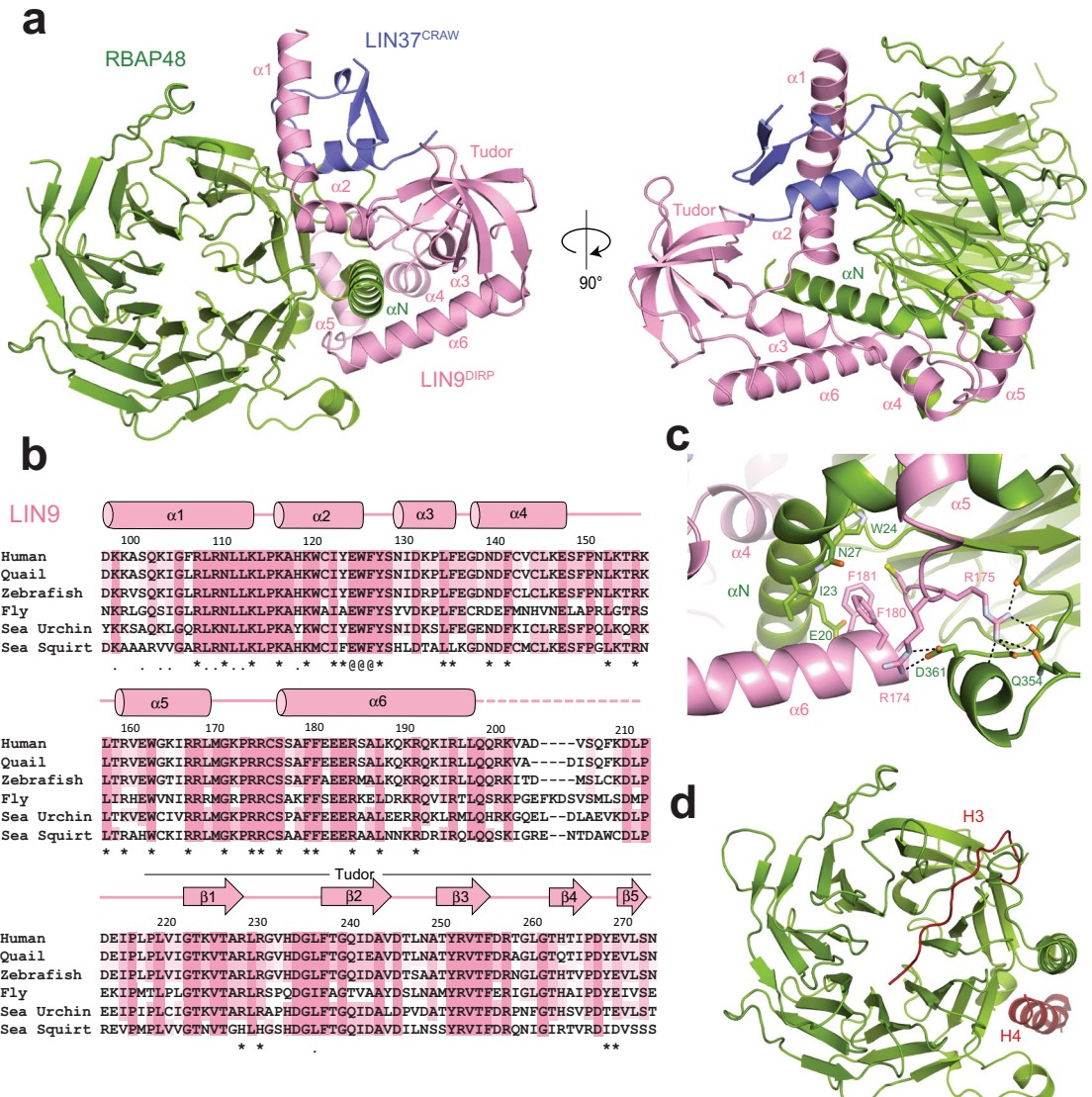

**Fig. 2 Structure of the MuvBN subcomplex. a** Overall structural model. **b** Alignment of LIN9 sequences from *H. sapiens*, *C. japonica*, *D. rerio*, *D. melanogaster*, *S. purpuratus*, and *C. intestinalis*. The (*) marks residues that contact RBAP48, the (.) marks residues that contact LIN37, and the (@) marks residues that contact both. **c** Close-up view of one interface between LIN9 and RBAP48. **d** Location of histone H3 and histone H4 peptide binding sites on RBAP48. When bound to LIN9, the H4 sites is blocked while the H3 site is mostly accessible. The model was generated from PDB IDs: 2YBA and 3CFV.

RBAP48 and LIN37. We conclude that the LIN9 N-terminus is necessary and sufficient for binding RBAP48 and LIN37, while the C-terminus binds LIN52 and LIN54 (Fig. 1b). We found that co-expression of RBAP48 with the DIRP region of LIN9 (LIN9[94–278]) and the conserved CRAW domain of LIN37 (LIN37[92–130]) in Sf9 cells yielded a MuvB subcomplex that was stable through affinity purification and size-exclusion chromatography (Supplementary Fig. 1). We call this subcomplex MuvBN, as it contains sequences toward the N-termini of LIN9 and LIN37.

**Overall structure of LIN9-LIN37-RBAP48 subcomplex.** We were able to crystalize the MuvBN subcomplex, and we determined the structure to 2.55 Å by molecular replacement using the known RBAP48 subunit structure as an initial model (PDB: 3GFC) (Supplementary Table 1)[46]. The crystal structure contained one complex in the asymmetric unit, and we built the LIN9 and LIN37 fragments into the unmodeled electron density. The final refined MuvBN model contains one copy of each protein (Fig. 2). As previously described, RBAP48 has a β-propeller domain fold,

consisting of seven small β-sheets, along with a single N-terminal helix[46–48]. The atomic structure of RBAP48 in MuvBN aligns well with other structures of the protein in other complexes with RMSDs ~0.3–0.6 Å (Supplementary Fig. 2). The LIN9[94–278] sequence is almost entirely visible in the electron density and contains six alpha helices and the Tudor domain. The helices are N-terminal to the Tudor domain and do not appear to form a globular structure. Instead, they wrap around and from extensive contacts with RBAP48, create a binding site for LIN37, and anchor the Tudor domain to the rest of the complex. The LIN37[92–130] CRAW domain is also nearly all visible in the electron density. This continuous LIN37 sequence forms two small β-strands and a short α helix. Our recombinant LIN9 was unstable without co-expression of this highly conserved fragment of LIN37[92–130], which interacts with both RBAP48 and LIN9 in the subcomplex.

**Structure of the LIN9-RBAP48 interface.** LIN9 and RBAP48 associate across a broad interface focused around the N-terminal helix (αN) of RBAP48 and the adjacent side of the β-propeller

domain (Fig. 2). All six LIN9 helices contact RBAP48, and five of them (α2–α6) surround and make interactions with the RBAP48 αN helix (Fig. 2a, b). Numerous hydrophobic, polar, and electrostatic contacts are observable between the proteins (Fig. 2b), and we highlight a few specific examples here that are relevant for the mutagenesis experiments described below. For example, a cluster of two arginines (R174 and R175) and two phenylalanines (F180 and F181) in LIN9 anchor α6 and the preceding loop against the RBAP48 αN helix and so-called PP-loop, which is an insertion in the sixth propeller β-sheet (Fig. 2c). The sidechains of R174 and R175 make a series of electrostatic interactions with side chain and main chain atoms in RBAP48 residues Q354, D358, P361, and G362, while F180 and F181 pack against RBAP48 residues I23 and W24.

Structures of RBAP48 and RBAP46 bound with various peptides depict how they are assembled into diverse complexes. A survey of known structures reveals two common peptide binding sites on the β-propeller domains (Fig. 2d and Supplementary Fig. 2). One site is across the face of the β-propeller and is found occupied by histone H3, Fog1, and PHF6. The second site is along the side of the propeller between αN and the PP loop; it is found occupied by histone H4, Mta1, and Suz12. In the MuvBN structure, the H3 site is for the most part accessible, although the α1 and α2 helices of LIN9 pack against the edge of the propeller where the H3 site-binding peptides exit the propeller face. In contrast, the H4 site is bound by the sequence in LIN9 between α5 and α6 and is not accessible in the MuvBN complex. It was recently reported that the proliferating cell nuclear antigen (PCNA)-associated factor (PAF) binds RBAP48 through a sequence in RBAP48 (residues 346–352) that in our structure is near the H4 site but somewhat solvent exposed[30]. It is feasible that PAF could access this extruded part of RBAP48 in the MuvB complex; however, how PAF binding to RBAP48 competes with p130 binding, as suggested[30], is unclear considering our result that MuvBN components are not required for p130 association (Fig. 1d).

Several structures of RBAP48 in complex with one or more proteins or larger protein fragments have also been previously determined. For example, RBAP48 is present in the polycomb complex PRC2[37,49]. As observed in MuvBN, RBAP48 is bound in these other complexes at multiple sites and on both sides of αN. One striking difference in how LIN9 and LIN37 bind RBAP48 compared to how proteins bind in other complexes is the extensive interactions with a glycine-rich loop in RBAP48 (residues 88–115) (Fig. 3a). This RBAP48 loop, which is an insertion between two strands in the first complete propeller blade, is disordered in almost all the structures with peptides and is partially ordered when binding Mta1 or the polycomb complex protein Suz12 (Supplementary Fig. 2). In contrast, the interactions of the insertion loop with LIN9 and LIN37 are much more extensive, and the entire loop appears ordered in the MuvBN structure. With respect to histone binding, the H4 binding site in RBAP48 is occluded and the H3 site is more accessible in both the PRC2 and MuvBN complexes.

**The LIN9 Tudor domain has a non-canonical aromatic cage.** LIN9 additionally contains a conserved Tudor domain that is visible in the subcomplex (residues 223–273). Tudor domains are protein interaction modules that are found in many chromatin-binding proteins. In several cases, they recognize methylated lysines and arginines and function as readers of modified histones[50–52]. The Tudor structure is defined by five anti-parallel β-strands that fold into a barrel. Target peptides are bound by an aromatic cage at one end of the barrel. The cage typically surrounds the modified basic side chain and makes stabilizing π-cation interactions. We aligned the LIN9 Tudor domain with structures of the PHF1 (PDB: 2M0O, RMSD 1.0 Å) and the SMN (PDB: 4A4E, RMSD 0.9 Å) Tudor domains in complex with their target peptides (Supplementary Fig. 3)[50,52]. The alignments suggest that the LIN9 cage contains fewer aromatics and is relatively inaccessible, as it makes an interaction with a loop that adjoins β3 and β4 at L261 (Supplementary Fig. 3). We note that we have not been able to detect binding of the LIN9 Tudor domain to several unmodified and modified histone peptides or modified lysine and arginine at high concentrations. While we do not rule out the possibility that the LIN9 Tudor domain binds histones or other proteins, we conclude that the structural features of the cage that mediate the interactions of other Tudor domains are not obviously present in LIN9.

**LIN37 structure and interface with LIN9 and RBAP48.** Previous functional domain mapping studies demonstrated that two highly conserved sequences in LIN37 were critical for LIN37 binding to other DREAM components and for DREAM repression of cell-cycle genes[24]. These sequences in LIN37 correspond with the CRAW domain of LIN37 that appears structured in our crystals of MuvBN, and they play a critical role in interacting with LIN9 and RBAP48. This observation firmly implicates the MuvBN subcomplex as the structural subunit of DREAM responsible for gene repression.

The small structured LIN37 CRAW domain is bound between LIN9 α1 and α2 (Fig. 3a–c). The two LIN9 helices form a V-shape that straddles one face of the LIN37 structure. Sidechains along one hydrophobic face of the LIN9 α1 helix (I104, L108, L111, and L112) are inserted into a groove formed by hydrophobic residues from all the LIN37 secondary structure elements (I97, L99, F100, V104, L106, F109, L115, I118, and W122). The LIN9 α2 helix binds the opposite face of the LIN37 helix from the LIN9 α1 helix with LIN9 W125 packing against the LIN37 backbone and interacting with LIN37 Y116. The LIN37 helix forms the primary interface between LIN37 and RBAP48 (Fig. 3a). Y116 and R120, both of which are highly conserved among LIN37 orthologs, make several interactions with the glycine-rich insertion loop in RBAP48. The nearby LIN9 α2 helix also contributes to this interface such that E124 from LIN9, Y116 from LIN37, and Y98 from RBAP48 all interact through a network of hydrogen bonds.

We tested the importance of several interface contacts observed in the structure on assembly of MuvB in HCT116 cells (Fig. 3d). We expressed either FLAG-tagged WT LIN9 or two FLAG-tagged LIN9 mutants and performed anti-FLAG immunoprecipitation to assay association with other MuvB proteins. A triple mutant (E125A/W126A/F127A) that contains mutations in the LIN9 α1 helix (LIN9[3X]) failed to co-precipitate LIN37, whereas a quadruple mutant (R174A/R175A/F180A/F181A, LIN9[4X]) with mutations in LIN9 α6 and the preceding linker (Fig. 2c) failed to co-precipitate both LIN37 and RBAP48. It is notable that LIN37 was lost in the LIN9[4X] co-precipitation even though the mutated residues are not directly at the LIN37 interface. These results indicate that despite the extensive interface, RBAP48 association with LIN9 can be disrupted through a few key mutations. The results of the LIN9[4x] mutant experiment also suggest that LIN37 association with LIN9 is likely stabilized by the presence of RBAP48 in the complex.

Analysis of the interactions at the LIN9-LIN37-RBAP48 interface reveals the structural mechanism for the specificity of RBAP48 in the MuvB complex. Previous analysis of MuvB components using mass spectrometry did not identify the presence of the RBAP48 paralog RBAP46[22]. In our co-immunoprecipitation experiments, we also did not observe association of RBAP46 with components of the complex (Fig. 3e).

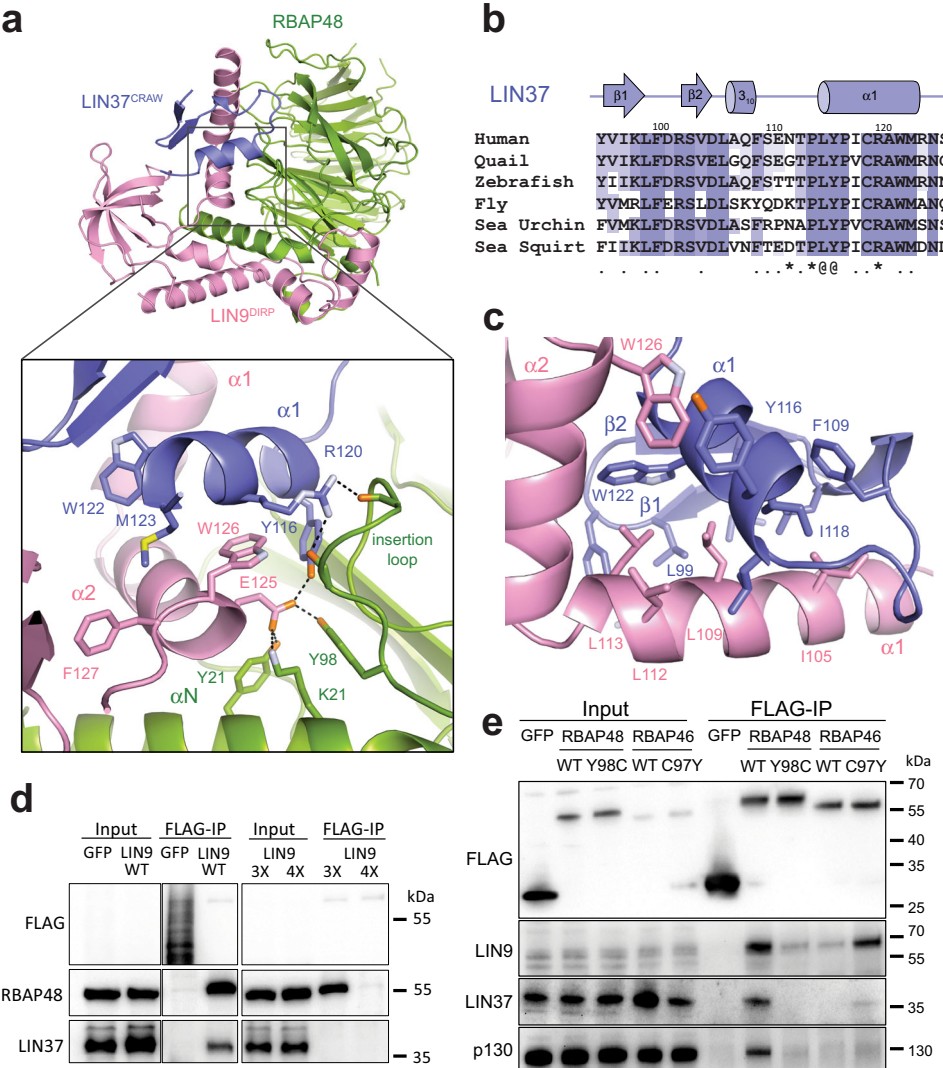

**Fig. 3 LIN37 CRAW domain binds both LIN9 and RBAP48. a** Interactions of LIN9 and LIN37 with the RBAP48 insertion loop. **b** Alignment of LIN37 sequences from organisms as in Fig. 2b. Residues that contact LIN9 (.), RBAP48 (*), and both LIN9 and RBAP48 (@) are indicated. **c** Close-up view of the LIN9-LIN37 interface. **d** The indicated FLAG-GFP control and FLAG-LIN9 WT and mutants were expressed by transient transfection in arrested HCT116 cells. Proteins were immunoprecipitated using anti-FLAG beads and the indicated proteins visualized by western blot. The 3x LIN9 mutant is E125A/W126A/F127A and the 4x LIN9 mutant is R174A/R175A/F180A/F181A. **e** Same as (**d**) but expressing the indicated RBAP46 and RBAP48 WT and mutant proteins. These immunoprecipitation experiments were performed each with a biological replicate, and results were similar.

The two human homologs are 89% identical, but notably, RBAP46 contains a cysteine at position Y98 in RBAP48. In the MuvBN structure, Y98 is in the RBAP48 insertion loop and is involved in a network of hydrogen bonds at the interface with both LIN9 and LIN37 (Fig. 3a). We found that while Flag-tagged wild-type mouse RBAP48 could co-precipitate MuvB components in HCT116 cells extracts, mouse RBAP48 with an RBAP46-mimicking Y98C mutation does not co-precipitate MuvB components (Fig. 3e). Conversely, a mouse RBAP48-mimicking C97Y mutation in mouse RBAP46 results in some additional affinity, although we note that the association still appears weaker than with WT RBAP48. We conclude that the MuvB complex has specificity for RBAP48 and that this specificity arises through this unique insertion loop association with LIN9 and LIN37.

**MuvB binds histone H3 tails and reconstituted nucleosomes lacking a CHR site.** The MuvB complex contains two domains that have potential histone binding properties: the Tudor domain of LIN9 and the β-propeller domain of RBAP48. We wanted to

test whether these domains, within the context of MuvB, are able to engage with histone peptides and nucleosomes. We first tested whether our recombinant purified MuvB complexes bind histone peptides that are known to form complexes with RBAP48. We tested binding of both MuvB (Fig. 4a) and the MuvBN sub-complex (Supplementary Fig. 4a) to fluorescein-labeled H3 (1–21) and H4 (21–41) peptides by fluorescence polarization. We found that MuvB and MuvBN bound the H3 tail but that they did not bind the H4 peptide. This observation is consistent with the MuvBN structure, which shows that the H3 site in RBAP48 is accessible while the H4 site is occluded by LIN9 (Fig. 2d). Using the fluorescence polarization assay, we found that MuvBN binds H3 peptide with similar but slightly weaker affinity as the full MuvB complex, suggesting that the MuvBN complex is sufficient to make the most significant contacts with the H3 peptide (Supplementary Fig. 4a). Isothermal titration calorimetry measurements also demonstrate binding of MuvBN to H3 but not H4 tails and suggest that H3 binding is mediated through RBAP48 as previously described (Supplementary Fig. 4b)[47,48].

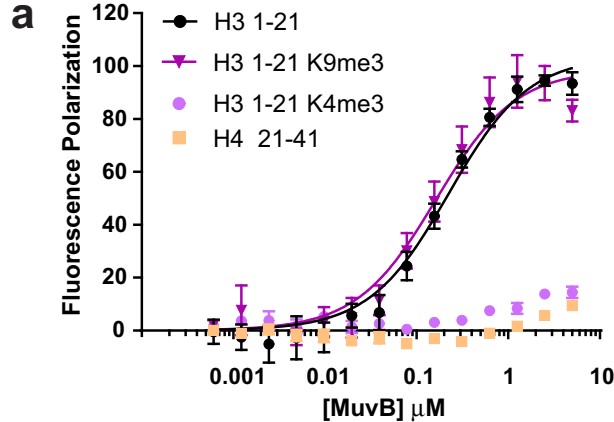

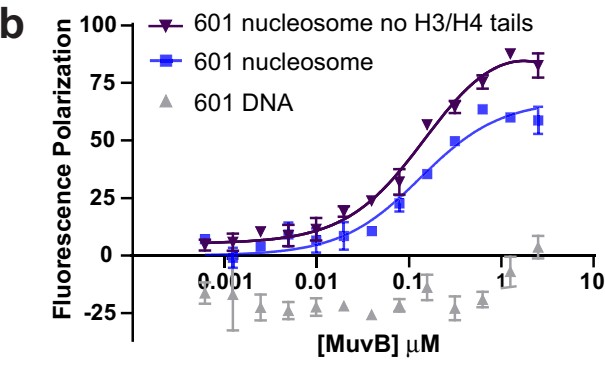

**c**

| | MuvB K$_d$ (nM) |
|---|---|
| H3 1-21 | 240 ± 30 |
| H3 1-21 K9me3 | 250 ± 40 |
| H3 1-21 K4me3 | no binding |
| H4 21-41 | no binding |
| 601 nucleosome | 130 ± 20 |
| 601 nucleosome no H3/H4 tails | 100 ± 10 |
| 601 DNA | no binding |

**Fig. 4 MuvB binds to histone peptides and nucleosomes.** Fluorescence polarization (FP) measurements of association between recombinant MuvB and dye-labeled histone peptides (**a**) or nucleosomes (**b**). The data are shown as mean values from three replicates with the standard deviation (SD) as error bars. Data are fit assuming a single binding constant. **c** Tabulated affinities and error estimates from global fitting of all data across three replicates. Source FP data are provided as Source data.

To probe whether posttranslational modifications on H3 tails influence MuvB binding, we tested two H3 marks that are associated with active transcription (H3K4me) or transcriptionally silent heterochromatin (H3K9me3). We found that MuvB bound H3 tails when methylated at K9 but failed to bind with H3 tails when methylated at K4 (Fig. 4a). This result is consistent with available structural data demonstrating that K4 methylation inhibits H3 tail binding to RBAP48[47]. We note that our observations contrast with experiments performed with purified *Drosophila* dREAM complex, which bound non-acetylated H4 peptides[20]. However, the Drosophila complex contains additional histone-interacting proteins (L3MBT and an HDAC ortholog) not present in the mammalian complex.

We then asked whether MuvB could bind reconstituted nucleosomes and whether nucleosome binding was conferred

by H3 tails alone. We reconstituted nucleosomes with full-length histones and the Widom 601 strong positioning sequence containing a fluorescein label. MuvB bound to these nucleosomes with slightly greater affinity than to the tails (Fig. 4b, c). The 601 DNA sequence lacks a CHR sequence, and we found that MuvB did not bind fluorescein-labeled free 601 DNA, indicating that nucleosome association occurs independently of DNA consensus motif binding. In the FP assay, we found that MuvB binds to the nucleosomes lacking histone H3 and H4 N-terminal tails (H3:39–136; H4:19–103) with a similar affinity compared to nucleosomes with tails (Fig. 4b). This observation is consistent with a known association of RBAP48 with tailless histone H3–H4 dimers, although we detect here association in the context of a reconstituted nucleosome[53]. Our data indicate that MuvB can bind nucleosomes through the H3 tails but that H3-tail binding is not necessary for MuvB-nucleosome association. To rule out any potential binding of the LIN9 Tudor domain, we reconstituted a mutant MuvB complex harboring LIN9 Tudor aromatic cage mutations (L230A/F238A/F256A/H264A) and found this mutant engages with Widom nucleosomes similar to the wild-type complex in the FP assay (Supplementary Fig. 4c). This result suggests that the LIN9 aromatic cage is not necessary for binding nucleosomes. Considering these results together, we propose that MuvB engages with H3 tails and/or the folded octamer to bind nucleosomes and that this association is primarily mediated by the MuvBN subcomplex including RBAP48.

**MuvB binds and stabilizes nucleosome occupancy on a reconstituted and chromatinized cell-cycle gene promoter.** We previously analyzed late cell-cycle genes in available ENCODE data sets and found that DREAM target gene promoters show a higher nucleosome density within the few hundred bases downstream from the transcription start site relative to genes that lack a CHR site and relative to constitutively expressed genes[31]. Following our observation here that MuvB binds nucleosomes in the absence of additional factors, we tested whether MuvB directly increases nucleosome occupancy on cell-cycle gene promoters. We cloned and amplified a minimal promoter from the human *TTK* gene, which is a late cell-cycle gene regulated by MuvB[54]. We folded a purified TTK-derived 461 bp DNA fragment with recombinant histone octamer in the presence and absence of MuvB. This promoter DNA fragment contains a single CHR located 187 bp from the 5′ end (Fig. 5a and Supplementary Fig. 5a). An electromobility shift assay demonstrated that MuvB was able to associate with the chromatinized promoter (Fig. 5a).

We then cross-linked our chromatinized samples and several control samples with trimethylpsoralen, digested protein, and performed metal-shadowing electron microscopy to assess nucleosome occupancy along the DNA molecules across our conditions (Fig. 5b, c)[55,56]. In these experiments, the presence of nucleosomes is inferred from the appearance of nucleosome-sized bubbles in the micrograph (Supplementary Fig. 5b). As expected, we observed nucleosomes in the samples folded with histone octamers prior to cross-linking but not in the sample that only contained free TTK DNA. When MuvB is present in the reconstitution reaction, we observe more molecules containing nucleosomes and an increase in the average number of nucleosomes per molecule (Fig. 5c). Furthermore, the distribution of inferred nucleosomes titrated with MuvB concentration. When a lower concentration of MuvB was present, we observed fewer nucleosomes per molecule relative to the high-concentration condition. We conclude that MuvB stabilizes nucleosomes in the synthetic TTK promoter, as MuvB increased nucleosome occupancy in the equilibrium established by the reconstitution reaction. We did not observe a significant change in the

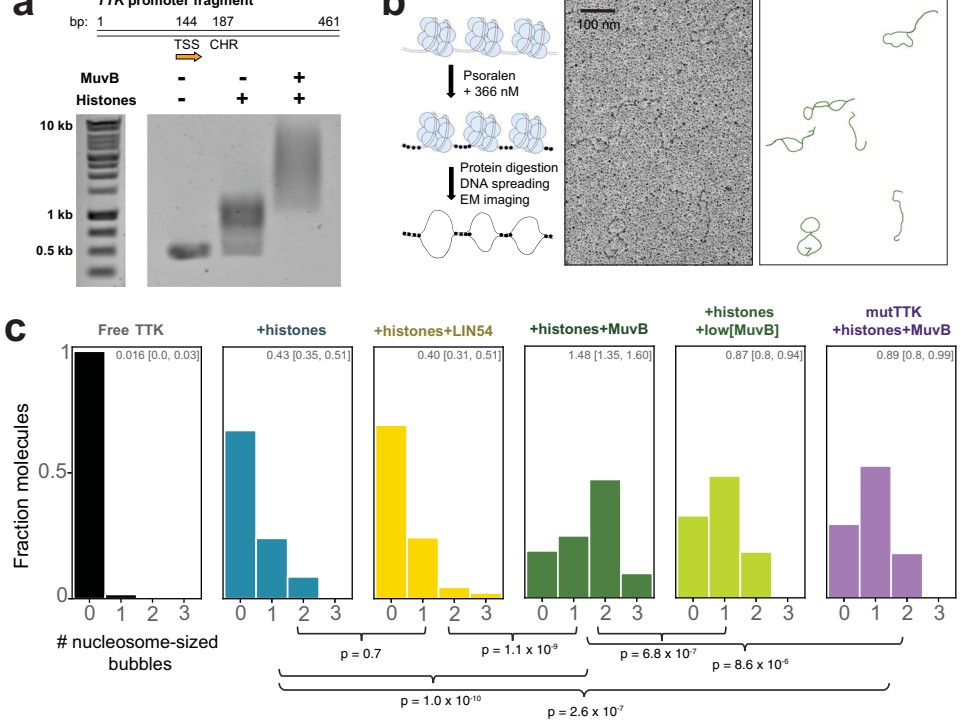

**Fig. 5 MuvB stabilizes nucleosomes on a reconstituted cell-cycle gene promoter. a** Reconstitution of the TTK promoter with nucleosomes and MuvB. At top is a schematic of the promoter fragment used in these experiments with the transcription start site (TSS) and CHR site indicated. The indicated proteins were refolded with a 461 bp fragment of DNA and the samples were analyzed by an agarose gel with ethidium bromide staining. The assay was replicated twice with similar results. **b** Schematic representation of the protocol for cross-linking and imaging (left) and example electron microscopy micrograph (middle) with corresponding traced molecules (right). **c** Histograms showing the fraction of DNA molecules containing the indicated number of nucleosome-sized bubbles for a sample of 100+ analyzed DNA molecules. The MuvB concentration used in the reaction was 1.3 μM, whereas low[MuvB] corresponds to 0.15 μM. The number at the top of the box indicates the average number of nucleosome-sized bubbles per DNA molecule with 95% confidence interval in brackets. These statistics were computed using bootstrapping with 10,000 iterations of resampling. Histograms were fit to a Poisson distribution and conditions were compared using an exact-Poisson test. *p*-values for comparisons of average number of bubbles (λ of Poisson distribution) across conditions are reported below the histogram.

nucleosome distribution with the inclusion of LIN54[504–709] alone, suggesting that binding of the CHR by the LIN54 DBD is not sufficient to increase nucleosome occupancy. When we mutated the CHR site in the TTK promoter, we observed a significant decrease in the average nucleosomes per molecule (Fig. 5c), which is consistent with weaker MuvB binding to the mutated CHR site in the DNA (Supplementary Fig. 5c). However, this average is still greater than the average in the absence of MuvB. We propose that MuvB binds and stabilizes nucleosome occupancy in the DNA even when LIN54 is not bound to the CHR site (i.e., in *trans* association with nucleosomes) but that simultaneous engagement of both the CHR and nucleosome (i.e., in *cis* association) results in increased stability. A histone chaperone-like activity has been reported for RBAP48 in other chromatin-bound complexes, and RBAP48 binds histone octamer intermediates[38,53]. Because our experiment probes the equilibrium established by the reconstitution reaction beginning with a folded octamer, we cannot rule out a role for MuvB in facilitating the assembly of nucleosome intermediates (Supplementary Fig. 5b). However, we favor the interpretation that, by binding the CHR and the nucleosome (Fig. 4b), MuvB stabilizes fully assembled nucleosomes in the promoter.

**MuvB associates with the +1 nucleosome in cell-cycle gene promoters**. We next used an MNase-ChIP approach to detect MuvB association with nucleosomes in cells (Fig. 6a)[57,58].

Chromatin preparations from HCT116 cells expressing Strep-tagged LIN9 were cross-linked and MNase digested. Samples were precipitated with Strep-Tactin, and after cross-links were reversed and protein was digested, DNA fragments were purified, ligated with barcoded adapters, and sequenced. In contrast to traditional ChIP experiments that identify transcription factor binding motifs, we aimed to purify nucleosomal-DNA fragments that associate with our transcription factor[57]. To enrich for these longer fragments (>100 bp), we ligated adapters after a SPRI-bead DNA purification. Compiled DNA sequences were aligned to the human genome, and we used MACS2 to locate enriched peaks corresponding to LIN9-interacting sequences.

We first analyzed precipitated DNA sequences from HCT116 cells that were treated with Nutlin-3a, which induces DREAM-mediated repression of both S phase and M phase genes through the p53 pathway (Supplementary Fig. 6)[45,59]. By comparing Strep-LIN9-precipitated samples to control samples in which cells were transfected with empty vector, we identified 253 genes with MACS peaks having >4.7-fold enrichment in sequencing reads. Gene ontology analysis of this data set reveals enrichment in genes related to cell cycle, mitotic division, and response to DNA damage (Fig. 6b). We found that 177 (70%) of the 253 most enriched genes have previously been identified as DREAM-regulated genes based on LIN9 and E2F4/p130 ChIP, RNA expression, and promoter analysis[15,22,34,45] (Fig. 6c). Moreover, these DREAM genes tended to show higher enrichment than the other identified genes among the top hits. We performed two

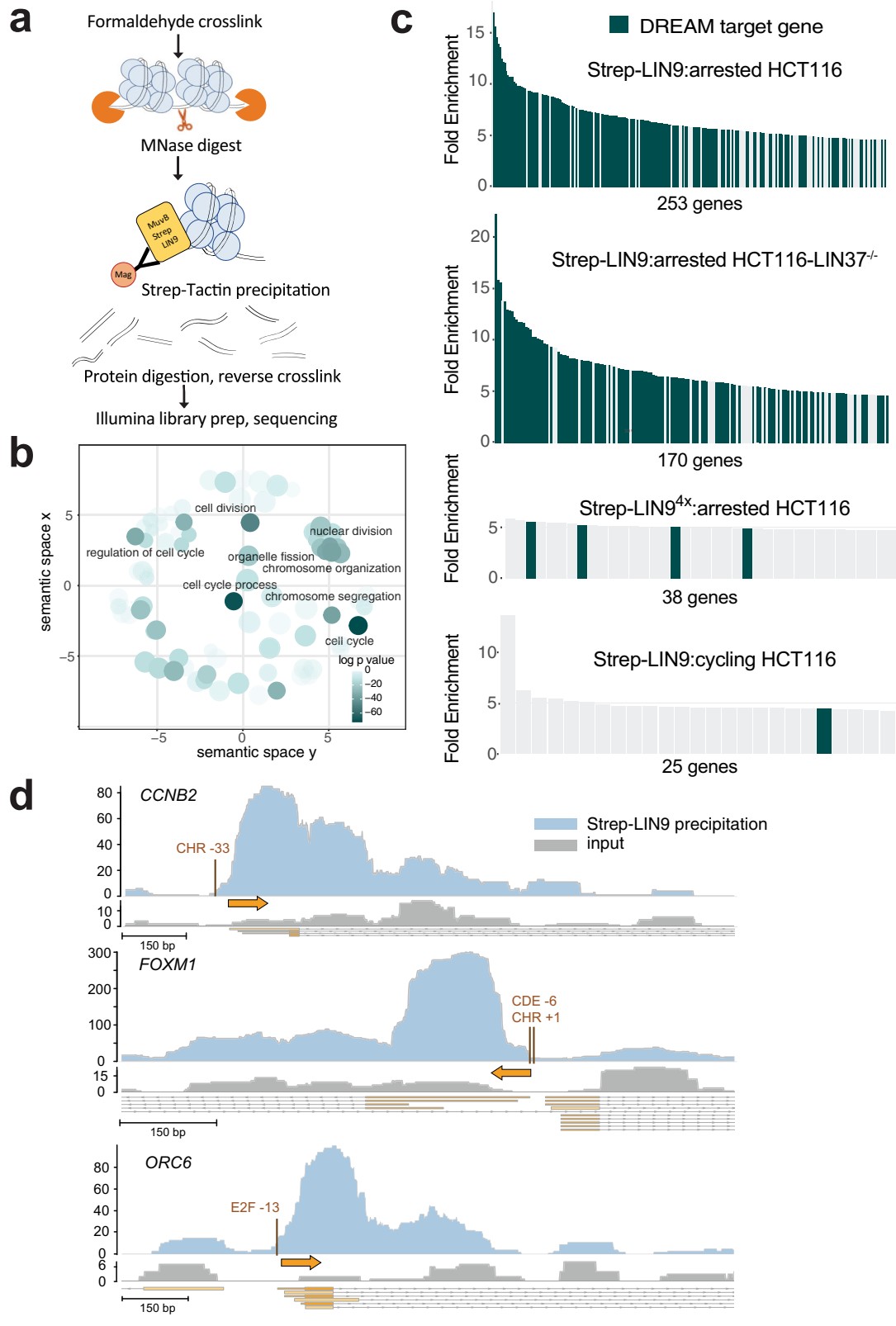

additional replicate experiments, one technical replicate with a different MNase concentration for the digestion and one biological replicate, and we found that the enrichment of many DREAM genes was reproducible (Supplementary Fig. 6b). We performed an analogous experiment in which we expressed Strep-

LIN9$^{4X}$. The LIN9$^{4x}$ mutant does not associate with LIN37 and RBAP48 but still associates with CHR consensus sites (Figs. 3d, 6c, and Supplementary Fig. 6a, c). Considering the same 4.7-fold threshold, this data set contained fewer genes overall and only four DREAM genes containing enriched sequences (Fig. 6c). We

**Fig. 6 MuvB associates with nucleosomes in DREAM-regulated gene promoters in arrested HCT116 cells. a** Diagram of the MNase-ChIP experiment designed to enrich nucleosome-sized DNA fragments that interact with MuvB complexes. **b** Gene ontology (GO) analysis of the enriched DNA sequences following MNase digestion and precipitation from arrested cell extracts. Gene groups with *p*-values < $1 \times 10^{-30}$ are labeled. A complete list of GO terms is provided as Source data. **c** Top enriched genes with DNA sequences that co-precipitated with Strep-LIN9. Experiments were performed using WT or LIN37$^{-/-}$ HCT116 cells, WT LIN9 or LIN9$^{4x}$, and in arrested or cycling cells. The number of genes with an enrichment >4.7-fold are indicated for each experiment. The list of enriched genes can be found on the NCBI GEO database (accession GSE189435). The annotated list of DREAM genes used as a cross-reference is provided as Source data. **d** Genome browser tracks corresponding to the *CCNB2*, *FOXM1*, and *ORC6* promoters. The number of DNA sequence reads is plotted for the input (gray) and Strep-LIN9 precipitated DNA samples. These data correspond to one replicate performed in arrested HCT116 cells. Data for other replicates and experiments are shown in Supplementary Fig. 6. The transcription start site (TSS) in each gene (base of orange arrow) along with the position of the DREAM-binding DNA motif relative to the TSS are indicated.

also performed an experiment precipitating Strep-LIN9 from extracts of cycling HCT116 cells and found enrichment of fewer genes compared to arrested cells (Fig. 6c). We conclude that our experimental protocol successfully enriches LIN9-bound DNA sequences at expected cell-cycle genes in arrested HCT116 cells and that enrichment depends on intact MuvBN.

Inspection of the WT LIN9-precipitated sequence reads aligned to the human genome reveals enrichment of DNA corresponding to nucleosome-sized fragments (~150 base pairs) near the transcription start site and E2F or CHR consensus sites in the DREAM-regulated genes (Fig. 6d and Supplementary Fig. 6d). For example, in the *CCNB2* promoter, which contains a canonical CHR DREAM-binding site, the strongest enriched peak is located just downstream of the closely spaced TSS and CHR site. This nucleosome corresponds to the +1 nucleosome, which has been previously identified as being well-positioned in repressed genes[60–63]. We observed secondary sites of enrichment, which correspond to nucleosomes (e.g., +2 and +3 nucleosomes) further downstream of the TSS. The enrichment decreases with increasing distance from the CHR site. In *FOXM1* and *ORC6*, which contain CDE-CHR and E2F binding sites for DREAM, respectively, we observed a similar pattern, with the +1 nucleosome showing the strongest enrichment, followed by a weaker coverage of the distal nucleosomes. Multiple lines of evidence suggest that these enriched fragments correspond to sequences in MuvB-bound nucleosomes rather than sequences protected simply by MuvB binding consensus DNA. First, the peaks are centered adjacent to MuvB-binding sites in DNA as opposed to centered on them, suggesting the read sequences are not protected from steric occlusion of MNAse by MuvB itself. Second, we do not see enrichment in the LIN9$^{4x}$ mutant or cycling cells experiments (Supplementary Fig. 6d), which probe conditions in which MuvB is still bound to chromatin and would still offer MNase protection. Third, secondary sites of enrichment even more distal to the CHR sites are also nucleosome-sized. Fourth, we commonly observed sequence reads of sizes corresponding to integral numbers of nucleosomes (Supplementary Fig. 6e).

We aligned the promoter regions of the 177 enriched DREAM genes according to their transcription start sites (TSS) to identify more broadly the structural signature of LIN9-associated nucleosomes. Most of these genes show a sharply positioned nucleosome within 150 bases downstream of the TSS (Fig. 7a). We conclude that LIN9 primarily precipitated the +1 nucleosome in these promoters. Considering that expressed wild-type LIN9 forms complexes with other endogenous MuvB components (Fig. 1d) but that LIN9$^{4X}$ does not associate with LIN37 and RBAP48 (Fig. 3d), we further conclude that these nucleosomes are bound by MuvB complexes and that these interactions are mediated by MuvBN.

We emphasize that the enriched nucleosomes in the set of DREAM genes do not overlap with the E2F and CHR consensus binding sites, suggesting that DREAM binds these DNA elements

in linker DNA and not in the nucleosome core particle. Nucleosomes are positioned next to the DREAM-binding site, which is typically in close proximity to the TSS (Fig. 6d), and do not necessarily contain the E2F and CHR DNA sequence motifs. We also note that the primary and secondary peaks in the sequence coverage persist when the reads are filtered for exclusively mononucleosome-sized inserts (Supplementary Fig. 6e). This observation that MuvB precipitated both proximal but not overlapping and distal nucleosomes to its consensus binding sequence further suggests that MuvB makes direct contact with the nucleosome core. This interpretation is consistent with our biochemical observations that interactions with nucleosomes are facilitated through protein-protein binding rather than through proximal DNA interactions (Fig. 5). Importantly, the peak corresponding to the +1 nucleosome is stronger and more tightly positioned in the precipitated sequencing data compared to the input data (Figs. 6d and 7a). This enrichment of a strongly positioned nucleosome is consistent with a role for MuvB in binding and stabilizing the +1 nucleosome in DREAM promoters.

**MuvB association with a tightly positioned +1 nucleosome correlates with gene repression.** We next performed a similar MNase-ChIP experiment using LIN37 knock-out HCT116 cells (HCT116-LIN37$^{-/-}$, Figs. 6c, 7a, b, and Supplementary Fig. 6a, d). In these cells, the MuvB complex assembles on CHR promoters, but cell-cycle genes are no longer fully repressed by DREAM when cells are arrested[24,45]. In fact, we observed in the input MNase data from the set of DREAM genes the nucleosome phasing pattern that is characteristic of genes undergoing transcription (Fig. 7b)[61,63]. We still observed enrichment of known DREAM genes in the pool of Strep-LIN9 precipitated DNA reads (Fig. 6c), and many of the enriched genes overlap between the data sets from wild-type and LIN37 knock-out cells (Supplementary Fig. 6b). We note that this result from precipitating Strep-LIN9 from knock-out cells is distinct from what we observed precipitating Strep-LIN9$^{4x}$ from wild-type cells. In the former experiment, LIN37 is missing from MuvB complexes, while in the latter both LIN37 and RBAP48 are missing; however, both complexes can associate with DNA (Fig. 3d and Supplementary Fig. 6C). From this comparison, we conclude that RBAP48 is necessary for nucleosome association.

Analysis of nucleosome occupancy generated for enriched genes in the data set from knock-out cells suggests that MuvB still associates with +1 nucleosomes (Fig. 7a, b, and Supplementary Fig. 7). However, the bound nucleosomes are distributed over a broader region of DNA, i.e., the boundaries of the positioned nucleosome are more poorly defined, and the position more typically encroaches on the TSS. We observe a significant (*p* < 0.05) difference in nucleosome occupancy comparing the wild-type and LIN37 knockout data sets in regions 100–200 bp both downstream and upstream of the TSS in DREAM genes (Supplementary Fig. 7b). We cannot determine that this broader

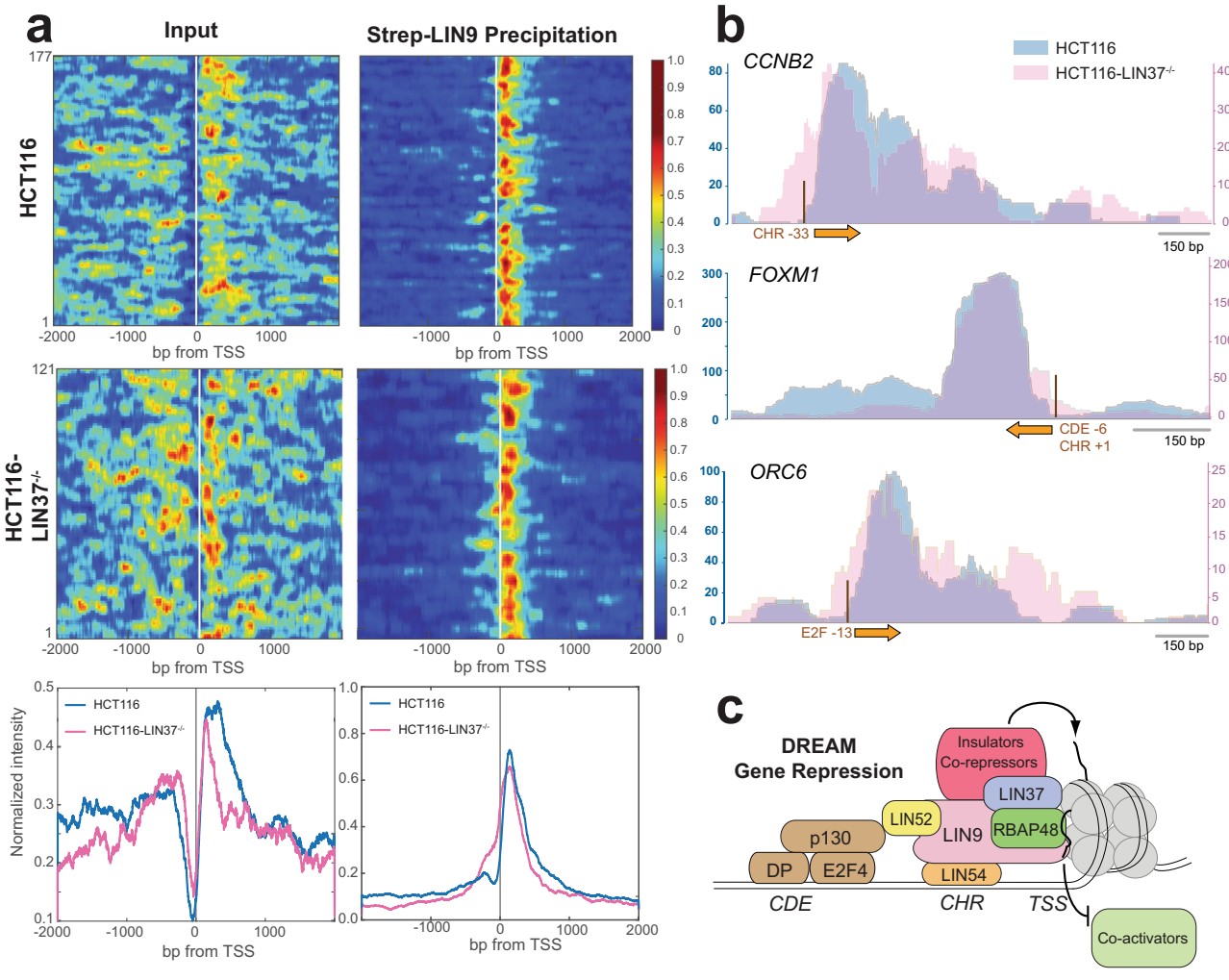

**Fig. 7 The sharp positioning of the MuvB-bound +1 nucleosome correlates with gene repression. a** Heatmaps of relative read density across all known DREAM genes that show >4.7-fold enrichment in the Strep-LIN9 precipitant and correspond. Both input and precipitant sequencing data are shown for the same gene set and in the same order. At the bottom are the aggregated and normalized read intensities across all genes shown in the heat map for the input and precipitated DNA data sets. Data from HCT116 (blue) and HCT116-LIN37$^{-/-}$ (pink) are shown. The observation of more peaks with periodicity corresponding to nucleosomes (i.e., the +2 and +3 nucleosome peaks) in the HCT116-LIN37$^{-/-}$ input data set is consistent with DREAM genes being active during Nutlin-3a-induced quiescence of those cells[45]. **b** Comparison of three example gene tracks showing the number of DNA sequence reads in Strep-LIN9 precipitated samples from HCT116 (blue, left axis) and HCT116-LIN37$^{-/-}$ (pink, right axis) cells. The TSS (orange arrow) and DREAM-binding DNA motifs are indicated. **c** Overall model showing organization of DREAM and the association between MuvBN and the +1 nucleosome, which we propose mediates gene repression.

distribution of nucleosome positions is directly a result of the absence of LIN37 from the complex or is a signature of expressed DREAM genes in the KO cells. Still, these results, together with our observation that Strep-LIN9 does not robustly precipitate nucleosomes from cycling cells, demonstrate that the sharply positioned MuvB-associated nucleosomes correlate with gene repression.

## Discussion

Genetic studies across model organisms all point to the function of the MuvB core as an intrinsically repressive complex that interacts with other TFs to modulate gene expression. In *C. elegans*, even when the p107/p130 ortholog is knocked out such that DREAM does not form on promoters, the MuvB core retains the ability to repress target genes[64]. In *Drosophila*, the lethal *myb-null* phenotype can be rescued by the loss of function of the fly orthologs of LIN9 and LIN37, which restores expression of MYB target genes[65–67]. In mammalian cells, LIN37 knockout or LIN9 knockdown leads to specific loss of repression of cell-cycle genes

upon driving cell-cycle exit[22,24,42], and a similar defect is observed upon loss of the RBAP48 ortholog in *Drosophila*[43]. While mammalian LIN9 loss also fails to activate mitotic genes, this activation defect may be linked to the requirement of LIN9 for recruiting B-MYB[22,35,41,42]. Together, these results demonstrate that LIN9, LIN37, and RBAP48 contribute to a repressive MuvB function.

Our results implicate the MuvBN subcomplex as the structural unit in MuvB responsible for this intrinsic repressive function and link repression to nucleosome binding. The structure and biochemical data demonstrate that LIN9 and LIN37 form a scaffold for the MuvB core in that they bind and assemble LIN52, LIN54, and RBAP48 (Fig. 7c). The MuvBN structure contains RBAP48 and conserved sequences of LIN37 that are both required specifically for cell-cycle gene repression. Our data demonstrate that MuvB binds and stabilizes nucleosomes and that MuvBN, which contains the repressor subunits, is sufficient for this interaction. We observed the association of LIN9-containing MuvB complexes with the +1 nucleosome in the

promoters of repressed cell-cycle genes in arrested cells, but this association is lost in cycling cells or with a LIN9 mutant that does not assemble the MuvBN components LIN37 and RBAP48. While we still see association of MuvB with nucleosomes in active gene promoters in arrested LIN37 knock-out cells, the associated nucleosome appears more strongly positioned under conditions of repression. We propose that MuvB stabilizes nucleosome position by making a bipartite interaction with the nucleosome and DNA. LIN54 binds the CHR sequence while MuvBN tethers the nucleosome near the CHR through a direct association with the histone tails or additional nucleosome contacts. We further propose that this association with +1 nucleosomes contributes to repression by inhibition of remodeling, polymerase activity, or posttranslational histone modification required for transcription (Fig. 7c). For example, association of MuvB with the histone H3 tail may sequester the tail from other chromatin-binding proteins and remodelers. Another possibility is that by tightly binding and positioning the +1 nucleosome, MuvB may increase the energy barrier that stalls RNA polymerase activity, resulting in uninitiated or aborted transcripts. By binding through multiple modes, i.e., histone tails and core, MuvB could prevent the unwrapping and movement of the +1 nucleosome. Although initial mass spectrometry analysis revealed few binding partners to human MuvB that could explain its repressive role, more recent studies have found MuvB can in certain contexts recruit proteins such as PAF and SIN3B[30,68]. Additional factors may also function to enhance repression in addition to the nucleosome binding activity of MuvB. Our result that MuvB can bind nucleosomes even in the absence of an H3 tail interaction suggests that the H3 site in RBAP48, which our structure shows is accessible in MuvB, might be used by MuvB to recruit other repressor complexes.

Research on the structure of chromatin has revealed important factors that determine the nucleosome position in the genome, including intrinsic properties of DNA sequence, chromatin remodeling complexes, the polymerase machinery, and sequence-specific TFs[1,3,4]. The role of TFs has focused on their potential for maintaining the nucleosome depleted region around the TSS and for establishing the +1 nucleosome. Evidence supports a model in which the mutually exclusive interaction between TFs and histones for DNA allows TFs to act as a barrier for nucleosome deposition such that the +1 and other proximal nucleosomes form at the closest accessible sites. Our data support a more direct function for TFs in establishing the +1 position through physical association and correlate this association in cells with a more tightly positioned nucleosome at repressed genes. Under conditions when MuvB is not actively repressing (LIN37 KO cells in quiescence), we observe more variability in the nucleosome position. The extent to which these observations result from the dynamics of RNA polymerase during the transition from repressed to active genes remains uncertain.

Several important questions remain about this MuvB repressive function including the structural mechanism of nucleosome recognition and the role of LIN37. RBAP48 in many studies is sufficient for nucleosome binding, and it is still present in MuvB complexes that lack LIN37 yet cannot repress gene expression[24]. We speculate that this non-functional complex may be unstable or improperly structured such that it cannot enact repression or bind co-repressors. The extensive interaction interface and co-dependence of their association in our mutagenesis study support the hypothesis that the core subunits of MuvBN, (LIN9, LIN37, and RBAP48) co-fold to form a stable complex. Another important remaining question is how the structure and function of MuvB changes such that it switches from a repressor to an activator of gene expression once cells enter the cell cycle. MuvB components are still present on the promoter and are required for

recruiting B-MYB and FOXM1. One possibility is that MuvB repression activity is relieved, for example by the binding of B-MYB, which is consistent with observations in *Drosophila* that the MuvB-binding sequence in B-MYB is alone sufficient to rescue a B-MYB deletion phenotype[69]. Another possibility is that Cdk phosphorylation, detected on all the MuvB subunits, plays a role in modulating MuvB function[70]. A third possibility is that in addition to its repressive function, MuvB positions the +1 nucleosome to prime genes for expression upon the binding of the activator transcription factors B-MYB and FOXM1. In this mechanism, MuvB may facilitate the acetylation of histones by the p300 acetylation machinery, which is recruited by the activator TFs. Through TF and p300 association, MuvB may also help recruit the basal transcription machinery. Finally, it will be important to understand how widespread interactions of TFs with the +1 nucleosome are and how these interactions regulate chromatin and gene expression

## Methods

**Plasmids for protein expression in mammalian cells.** The LIN9 and RBAP48 ORFs were amplified from cDNA derived from mouse NIH3T3 cells by standard PCR. The EGFP ORF was amplified from pEGFP-N1 (Clontech). The ORFs were cloned into pcDNA3.1(+) and fused either with an N-terminal 3xFlag tag (LIN9) or an N-terminal 1xFlag tag (RBAP48, EGFP). Site-directed mutagenesis was performed following the QuikChange protocol (Stratagene). For MNase-Seq experiments, the LIN9 ORFs were subcloned into pcDNA3.1(+) containing an N-terminal Twin-StrepII tag.

**Recombinant protein expression and purification.** To assemble the entire MuvB complex, proteins (GST- or Strep-LIN9[94–542], GST-LIN37, GST-LIN52, His- or GST-LIN54[504–749], and Strep-RBAP48) were co-expressed in Sf9 cells via baculovirus infection. Cell pellets were harvested after 72 h of growth in suspension at 27 °C, and complexes were purified using GST-affinity purification followed by Strep-affinity purification. After removal of affinity tags through TEV protease cleavage, purified complexes were isolated through size-exclusion chromatography using a Superdex 200 column. The final buffer contained 200 mM NaCl, 25 mM Tris HCl, and 1 mM DTT at pH 8.0. The MuvBN subcomplex was assembled by co-expressing GST-LIN9[94–278], LIN37[2–130], and full-length RBAP48 in Sf9 cells as described for the full complex. The subcomplex was purified using GST-affinity purification followed by anion exchange. Affinity tags were then removed with TEV protease and the complex was isolated with a Superdex 200 column. The final buffer contained 150 mM NaCl, 25 mM Tris HCl, and 1 mM DTT at pH 8.0.

**X-ray crystallography.** The MuvBN subcomplex was crystallized in a sitting drop at 4 °C containing 0.2 M sodium tartrate tetrahydrate, 0.1 M bis-tris propane pH 6.5, and 20% PEG 3350. Crystals were harvested and directly frozen in liquid nitrogen. Data were collected at $\lambda = 1.0332$ Å and 100 K on Beamline 23-ID-B at the Advanced Photon Source, Argonne National Laboratory. Diffraction spots were integrated with Mosflm[71]. Phases were solved by molecular replacement with PHASER[72] and using RBAP48 (PDB: 3GFC) as a search model. The initial model was rebuilt with Coot[73], and LIN9 and LIN37 were added to the unmodeled electron density. The resulting model was refined with Phenix[74]. Several rounds of position refinement with simulated annealing and individual temperature-factor refinement with default restraints were applied. The final refined model was deposited in the Protein Data Bank under Accession Code PDB ID: 7N40.

**Co-immunoprecipitation and DNA affinity experiments.** Human HCT116 colon carcinoma cells were cultivated in DMEM supplemented with 10% FBS. Transfections were performed in 10 cm plates using 7 µg plasmid and 35 µl PEI per plate. To stimulate DREAM formation, cells were treated with 10 µM Nutlin-3a for 24 h. Cells were harvested 48 h after transfection. Whole cell extracts were prepared by lysing the cells in IP lysis buffer (50 mM TRIS-HCl pH 8.0, 0.5% Triton-X 100, 0.5 mM EDTA, 150 mM NaCl, 1 mM DTT, and protease inhibitors) for 10 min on ice followed by 5x 1 s direct sonication. Flag-tagged proteins were immunoprecipitated from 2 to 3 mg cellular extracts with Pierce Anti-DYKDDDDK Magnetic Agarose (Invitrogen). Beads were washed 5x with 1 ml IP lysis buffer end eluted with 50 µl 1xLaemmli buffer. Twelve micrograms of input samples and 12 µl IP samples were analyzed by SDS-PAGE and western blot following standard protocols. The following antibodies were applied for protein detection: FLAG-HRP (RRID:AB_2017593, Santa Cruz Biotechnology; dilution 1:2000), p130/RBL2 (D9T7M) (RRID:AB_2798274, Cell Signaling; dilution 1:1000), LIN54 A303-799A (RRID:AB_11218173, Bethyl Laboratories; dilution 1:1000), LIN9 ab62329 (RRID:AB_1269309, Abcam; dilution 1:1000), RBBP4 A301-206A (RRID:AB_890631, Bethyl Laboratories; dilution 1:5000), LIN37-T3 (custom-made at Pineda Antikörper-Service, Berlin, Germany; dilution 1:1000)[54].

For DNA affinity purifications, HCT116 cells were cultivated in 15 cm plates and transfected with 70 µl PEI and 15 µg plasmids expressing wild-type and mutant LIN9 fused with an N-terminal 3xFlag tag. Twenty-four hours after the transfection cells were treated with 5 µM Nutlin-3a for 48 h. Affinity purifications were performed as described earlier[75]. Biotinylated DNA probes were either amplified from the pGL4.10 empty vector or from pGL4.10 containing the mouse *Ccnb2* CDE/CHR MuvB-binding site[34]. The following antibodies were applied for protein detection: FLAG-M2 (RRID:AB_262044, Sigma-Aldrich; dilution 1:1000), p130/RBL2 (D9T7M) (RRID:AB_2798274, Cell Signaling; dilution 1:1000), LIN37-T3 (custom-made at Pineda Antikörper-Service, Berlin, Germany)[54], Histone H3 (RRID:AB_331563, Cell Signaling Technology; dilution 1:1000).

**Nucleosome reconstitution**. Xenopus histones as well as their tailless counterparts were expressed and purified in *E. coli* as inclusion body preparations as previously described[76,77]. Octamer reconstitution was completed by mixing equi-molar amounts of purified histones in a buffer containing 7 M guanidinium HCl, 20 mM Tris pH 7.5, and 10 mM β-mercaptoethanol, followed by dialysis into 2 M NaCl, 10 mM Tris pH 7.5, 1 mM EDTA, and 5 mM β-mercaptoethanol. Folded octamers were purified using size-exclusion chromatography on a Superdex 200 column. Nucleosome reconstitution was performed by mixing purified histone octamers with the Widom 601 positioning sequence and de-salting by gradient dialysis[76]. For Widom nucleosomes, we used a 1.1:1 ratio of octamers:DNA molecules. At a salt concentration of 50 mM, nucleosome samples were collected in a buffer containing 50 mM Tris pH 7.5 and 1 mM DTT.

**Fluorescence polarization assay**. Histone peptides were synthesized with fluorescein. For experiments with Widom nucleosomes, the 601 sequences were PCR amplified with a primer containing fluorescein and reconstituted with octamer as described above. 20 nM peptide was mixed with varying concentrations of MuvB protein complex in a buffer containing 50 mM Tris pH 7.5, 150 mM NaCl, 1 mM DTT, and 0.1% (v/v) Tween-20. Twenty microliters of the reaction were used for the measurement in a 384-well plate. Fluorescence polarization (FP) measurements were made in triplicate, using a Perkin-Elmer EnVision plate reader. The $K_d$ values were calculated using global fitting in Prism 8 (Version 8.2.1).

**Electron microscopy on reconstituted promoters**. The minimal region of the human *TTK* promoter (461 bp) was cloned, amplified by PCR, and purified by agarose gel extraction. Histone octamers were folded with the *TTK* DNA as described above for Widom nucleosomes, but we used an octamer to DNA ratio of 3.1:1 to allow for the formation of di- and tri-nucleosome species. For the relevant conditions, purified MuvB complex or LIN54 was added to the nucleosome folding reaction during the de-salting process at a NaCl concentration of ~800 mM. Cross-linking of gene promoters and electron micrograph preparation was performed as previously described[55]. In brief, samples were treated with trimethylpsoralen and UV radiation to allow double-stranded DNA cross-links to form at unprotected, octamer-free regions. Following cross-linking, proteins were digested by Proteinase K, and DNA molecules were purified, denatured, and spread across the surface of a copper transmission electron microscopy grid. Electron micrographs of all samples were prepared by rotary metal shadowing, and grids were visualized and collected on a JEOL 1230 TEM at the UC Santa Cruz IBSC Microscopy facility and a Tecnai 12 TEM at the UC Berkeley ELM lab. DNA molecules were traced, and molecular coordinates were saved using Fiji tools in the ImageJ software package as previously described[55]. The resulting traces were analyzed using custom python tools. Each DNA strand was traced such that an "end" of the molecule could be identified. Thus, every coordinate in one strand can be aligned to its complement by closest distance. Coordinates are assigned to base positions using a scale derived from the physical distance between coordinates within each strand and the known length of the TTK promoter (461 bp). A base pair is labeled single-stranded if the distance between strands exceeds a threshold distance, determined empirically. Once all base pairs are labeled, "bubbles" are determined by contiguous single-stranded stretches. Finally, a single-stranded "bubble" is labeled a nucleosome if its length is >90 base pairs (Supplementary Fig. 5). Bubble fusions occur such that two or more adjacent nucleosomes form one contiguous bubble; for this analysis, bubble fusions were labeled as a single nucleosome. This estimate was used because the number of total bubble fusions observed within the data set was small <5%.

**MNase-ChIP**. HCT116 cells were transfected with Strep-LIN9 constructs or an empty Strep expression plasmid. Twenty-four hours after transfection, cells were treated with 10 µM Nutlin-3a (Selleckchem) and harvested after 48 h. Cells were cross-linked with 1% formaldehyde for 15 min. The cross-linking reaction was stopped with glycine, cells were washed twice with PBS, and pellets were collected. Cell lysis and MNase digestion (1x or 5x) were performed as described earlier[58], and following digestion, LIN9-bound samples were precipitated using Streptactin-XT magnetic beads (IBA Lifesciences). Both input and IP samples were subject to RNAse treatment and proteinase K digestion and were reverse cross-linked by incubation at 65 °C for 16 h. DNA was purified by 2x SPRI bead clean-up. Library prep was carried out using NEB Next Ultra II kits, and paired-end sequencing was carried out on the NovaSeq 6000 platform with 150 bp paired-end mode for Illumina at Novogene Biotech, Co., LTD.

Sequencing reads were aligned against hg38 using the bwa-mem aligner[78,79]. Samtools and bedtools were used to convert data into bam and bed files, respectively. Peak calling for the precipitated samples was performed using the MACS2 -*bampe* algorithm and using the empty Strep-IP conditions as the control. To retrieve gene names for MACS2 peaks, coordinates were intersected with known genes using the Table Browser tool provided by the UCSC genome browser. Gene ontology analysis was performed on MACS2 peaks showing a >4.7-fold enrichment using the web-based tools GeneOntology.org and Revigo[80]. We generated coverage plots of our reads using Gviz and rtracklayer and other opensource R tools.

We utilized NucTools in paired-end mode to analyze nucleosome occupancy on input and Strep-LIN9 precipitated reads with single base-pair resolution (bin width = 1 bp)[81]. We restricted our analysis to genes that showed a MACS enrichment of >4.7-fold and were previously annotated to bind DREAM, to respond to p53 stimulation, and to become derepressed in LIN37 knockout cells[15,22,34,45]. We retrieved the TSS for this set of genes, either from those annotations or using bioMart[82]. As needed, the TSS sites were mapped on to hg38 using liftover. We oriented the output to center on the TSS and maintain a uniform direction of transcription. We then utilized the Cluster Map Builder feature of NucTools to generate aggregate plots and heatmaps of our genes.

**Reporting summary**. Further information on research design is available in the Nature Research Reporting Summary linked to this article.

## Data availability

The data that support this study are available from the corresponding authors upon reasonable request. X-ray diffraction data and model coordinates for the MuvBN structure in this study have been deposited in the Protein Data Bank under accession code 7N40. MNase-ChIP data have been deposited in the NCBI GEO database under accession code GSE189435. Source data are provided with this paper.

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

## Acknowledgements

This work was supported by a grant from the National Institutes of Health to S.M.R. (R01GM124148). GM/CA@APS has been funded by the National Cancer Institute (ACB-12002) and the National Institute of General Medical Sciences (AGM-12006, P30GM138396). This research used resources of the Advanced Photon Source, a U.S. Department of Energy (DOE) Office of Science User Facility operated for the DOE Office of Science by Argonne National Laboratory under Contract No. DE-AC02-06CH11357. The Eiger 16M detector at GM/CA-XSD was funded by NIH grant S10 OD012289. We thank the staff at the University of California Berkeley Electron Microscope Laboratory for advice and assistance in electron microscopy sample preparation and data collection. We thank Joseph Lipsick and Geeta Narlikar for histone plasmids.

## Author contributions

A.A., H.B., G.A.M., and S.M.R. conceived of and designed aspects of the study. A.A., P.R., A.H., K.Z.G., T.U.W., H.N., S.T., and G.A.M. acquired the data. A.A., P.R., A.H., K.Z.G., T.U.W., R.S., M.J.D., H.B., S.T., G.A.M., and S.M.R. analyzed and interpreted the data. A.A., G.A.M., and S.M.R. wrote the original draft of the manuscript. A.A., K.Z.G, R.S., M.J.D., H.B., G.A.M., and S.M.R. reviewed and edited the manuscript.

## Competing interests

The authors declare no competing interests.
