## [Peer Review File · Nature Communications]

REVIEWER COMMENTS

Reviewer #1 (Remarks to the Author):

The manuscript by Asthana and colleagues provides new structural information on the mammalian MuvB complex. The MuvB complex contains at least 5 proteins including LIN9, LIN37, LIN52, LIN54, and RBAP48 (RBBP4). These authors have previously reported structural studies on the interaction of LIN9 with LIN52 and B-Myb and between LIN52 and p107. In the current study, they focus on the N-terminus of LIN9 bound to LIN37 and RBBP4. They demonstrate that a structure containing the complete form of RBBP4, the conserved DIRP domain of LIN9, and a short conserved region (CRAW domain) of LIN37 formed a stable complex that yielded crystals. The authors demonstrate a specific interaction of LIN9/37 with RBBP4 but not the highly similar RBBP7 and generated a gain-of-function binding with RBBP7 with a point substitution that renders a residue critical for binding to LIN9 similar to that found in RBBP4. They find that the interaction of RBBP4, in complex with LIN9/LIN37, with histones is distinct from other structures of RBBP4 in different complexes. Importantly, they observe that the LIN9/LIN37/RBBP4 complex can increase the affinity of DNA for nucleosomes. In addition, they observed that the MuvB complex in cells appears to more efficiently protect the nucleosome positioned at +1 to the TSS. Given these observations, they propose that the LIN9/LIN37/RBBP4 complex contributes to gene repression by enriching for bound nucleosomes at the +1 position. They suggest that the ability of the MuvB complex to enhance the interaction of nucleosomes near the TSS provides a new model for transcription factor repression.

An interesting feature of the structural analysis that the LIN9 TUDOR domain does not interact specifically with histones or modified histones in contrast to the TUDOR domains of other proteins (PHF1 and SMN). Furthermore, RBBP4 selectively bound to Histone 3 tails but not Histone 4 tail residues. Notably, there was an increased affinity for binding to H3K9me3 peptides but no specific binding to H3K4me3 consistent with the ability to bind to repressed but not active promoters.

The authors contrast the ChIP performed with the LIN9 mutant that fails to bind to LIN37 and RBBP4 but retains binding to promoters with CHR sites and this failed to enrich for +1 nucleosomes. In contrast with ChIP performed in LIN37^{-/-} knockout cells, there was not an enrichment for +1 nucleosomes. Remarkably, the LIN37^{-/-} KO cells show very modest differences, perhaps highlighted by loss of a sharp demarcation at CDE/CHR sites. In contrast, the LIN94x mutant had weakened binding to DREAM targets. It is expected and would be helpful to show that the nucleosomes were not protected in the ChIP-seq analysis. This would support their model.

A recent study on the MuvB complex reported specific binding of PAF (also known as PCLAF/KIAA0101) to RBBP4 within the MuvB complex. Relevant to the current study, the earlier report mapped binding of PAF to an extruded region (S346-E352 AA) of RBBP4. The current study should at least comment on the possibility of these residues in RBBP4 being in a position to allow for PAF and histone binding in context of MuvB (Kim et al., 2021, *Molecular Cell* 81, 1698–1714 <https://doi.org/10.1016/j.molcel.2021.02.001>).

It would be helpful to have a map of the 461bp DNA minimal promoter from the human TTK gene, showing the single CHR site and TTS in Figure 5.

Reviewer #2 (Remarks to the Author):

In this manuscript, Asthana and colleagues investigate the assembly and chromatin localization of the essential but poorly characterized MuvB complex. The authors report the crystal structure of a subcomplex, consisting of the RBAP48, LIN9 and LIN37 subunits and describe the mechanism by

which the MuvB complex binds to the nucleosome. The authors also show that binding of the complex leads to an increase in nucleosome occupancy and that MuvB associates primarily with the +1 nucleosome. Using a combination of crystallographic, electron microscopy and MNase-ChIP approaches the authors generated data at molecular and atomic resolution level which allowed them to propose a model for the MuvB complex assembly at chromatin and explain the mechanism of target (cell cycle) gene repression by this complex. Overall, the manuscript describes novel findings, contains excellent quality data, and has justifiable conclusions.

A few comments:

1. It remains somewhat unclear as to how the complex stabilizes nucleosomes in the absence of other factors. The authors propose that MuvB could prevent the unwrapping and movement of the nucleosome. FRET assays with a fluorescently labeled nucleosome might help to clarify this.
2. Data shown in Fig. 4 were collected on the MuvB complex, which may complicate analysis of fluorescence curves, particularly in case if there are multiple binding sites and therefore multiple contacts, even weak, involved. It will be clearer and easier to understand contribution of each subunit to the binding through measuring binding affinities of isolated subunits and then comparing with the affinity of the entire complex.
3. Related to comment 2, if the association of the complex with the nucleosome is driven primarily by binding of RBAP48, it will be interesting to compare affinities of RBAP48 (published and reported here) for different ligands.
4. A minor point- the H3K9me3 is not a mark of transcriptional repression.

Reviewer #3 (Remarks to the Author):

The manuscript by Asthana et al described a molecular analysis of the MuvB complex, notable describing the molecular architecture of the complex and present a crystal structure of a subcomplex of MuvB critical for its function in repression of gene expression. MuvB along with Rb paralogs p130 or p177 and E2F4-Dp represses cell cycle genes in G0 or G1. On entry in to the cell cycle MuvB switches binding proteins to B-Myb and FoxM1 where it activates G2/M genes. MuvB is a 5 protein complex Lin9, Lin37, Lin52, Lin54 and RBAP48. A variety of additional experiments are presented to support a model where MuvB binds and stabilizes nucleosomes just downstream of the TSS on its target. Improper activation/repression by MuvB is associated with several cancers. Molecular details of MuvB assembly and molecular mechanism are poorly understood. The paper by Asthana et al does in my opinion fill a knowledge gap on the subject of MuvB complex architecture and mechanism of action. IN particular, some light is shed on the function of RBAP48 within the complex where it is proposed to function as a nucleosome interacting protein. The paper is well written, focusing on how MuvB functions to repress gene expression. I have only minor comments.

1. In the introduction it is noted that MuvB is conserved through animals but recent papers suggest that is more widely conserved albeit poorly characterized in protists (ciliates) as well (PMID: 34086947) which is probably worth noting.
2. In co-IP experiments of Fig1 why only express a small portion of Lin54?
3. Fig1C a little difficult to follow may help if lanes were numbered and then referred to by lane number in the text
4. Should the co-IP of MuvBN subunits (Supp Fig1) be moved to main text since this is the complex crystallized ?
5. There is a small band reacting with the flag antibody in the IP lane Fig1D Lin9 WT that I do not think is accounted for although I could be mistaken
6. RBAP48 is somewhat enigmatic with respect to function in diverse chromatin complexes - How do the histone H3 and H4 binding surfaces of RBAP48 compare when present in PRC2 to that of muvBN i.e. H4 occluded and H3 available - if known, how about for CAF-1 ?
7. Does the mutTTK site prevent or change muvB from binding in gel shift assay? i.e. this could

correlate with the decreased nucleosomes on the promoter

8. Can the authors comment on the difference in strategies of using WT Lin9 MNase ChIP compared to that of the Lin9 4x mutant that does not bind Lin37 and RBAP48 with performing Lin9 MNase ChIP +/- RNAi knockdown of RBAP48

9. Figure 7A, bottom panel left; since authors are measuring the difference in overall binding density around TSS (+1 nucleosome), I would suggest that authors perform a statistical test (for example, Mann-Whitney U tests) and calculate p-value (optional). I leave that to authors whether to calculate the significance (p-value) or not in this case, however.

10. Ciliates lack Lin37 as indicated in a recent report (PMID: 34086947, see above). Authors might want to mention this. Discussion section will benefit from this as it will highlight the Lin37 role in multicellular eukaryotes.

We thank the reviewers and editors for their appreciation of our study and for their insightful comments. We summarize below how the concerns and suggestions were addressed in a revised manuscript, including through addition of supplemental experiments and analysis.

Reviewer 1

1) The authors contrast the ChIP performed with the LIN9 mutant that fails to bind to LIN37 and RBBP4 but retains binding to promoters with CHR sites and this failed to enrich for +1 nucleosomes. In contrast with ChIP performed in LIN37^{-/-} knockout cells, there was not an enrichment for +1 nucleosomes. Remarkably, the LIN37^{-/-} KO cells show very modest differences, perhaps highlighted by loss of a sharp demarcation at CDE/CHR sites. In contrast, the LIN9^{4x} mutant had weakened binding to DREAM targets. It is expected and would be helpful to show that the nucleosomes were not protected in the ChIP-seq analysis. This would support their model.

Our analysis in Fig. 6 is not a traditional ChIP-seq experiment. We detected DNA sequences that were precipitated with strep-LIN9 following MNase digestion and enrichment of nucleosomal sized-DNA fragments. We therefore interpret the reviewer's comment to mean that it would be helpful to show representative genome browser tracks with enriched sequences identified in the LIN9^{4x} mutant precipitation. In the case of the LIN9^{4x} mutant, which does not associate with RBAP48 or LIN37, we only observed significant enrichment in the precipitates of a relatively small number of genes compared to wild-type LIN9 (Fig. 6C). We do show a representative gene for this experiment, *CCNB2*, in Supplementary Fig. 6D. The signal for the LIN9^{4x} mutant is weak and does not suggest interaction with the +1 nucleosome. We interpret these results as a lack of interaction between MuvB complexes containing the mutant LIN9 and nucleosomes.

In order to further support our model, we have added an additional MNase-ChIP experiment to Fig. 6C and Supplementary Fig. 6D in which Strep-LIN9 complexes are precipitated from extracts of cycling cells. The data from this experiment resemble the LIN9^{4x} mutant experiment, suggesting that MuvB-nucleosome interactions are not present in cycling cells and that nucleosome binding is associated with MuvB function specifically in gene repression during cell-cycle exit.

2) A recent study on the MuvB complex reported specific binding of PAF (also known as PCLAF/KIAA0101) to RBBP4 within the MuvB complex. Relevant to the current study, the earlier report mapped binding of PAF to an extruded region (S346-E352 AA) of RBBP4. The current study should at least comment on the possibility of these residues in RBBP4 being in a position to allow for PAF and histone binding in context of MuvB (Kim et al., 2021, Molecular Cell 81, 1698–1714 <https://doi.org/10.1016/j.molcel.2021.02.001>).

As suggested, we have made a comment in the revised manuscript reflecting on how our data fits with this PAF study. Interestingly, while our structure supports the idea that PAF can associate with RBAP48 in the context of MuvB, our other data on MuvB and DREAM assembly is not quite consistent with the proposed model that PAF binding to RBAP48 competes off p130. We added the following sentences:

It was recently reported that the proliferating cell nuclear antigen (PCNA)-associated factor (PAF) binds RBAP48 through a sequence in RBAP48 (residues 346-352) that in our structure is near the H4 site but somewhat solvent exposed³⁰. It is feasible that PAF could access this extruded part of RBAP48 in the MuvB complex; however, how PAF binding to RBAP48 competes with p130 binding, as suggested³⁰, is unclear considering our result that MuvBN components are not required for p130 association (Fig. 1D).

3) *It would be helpful to have a map of the 461bp DNA minimal promoter from the human TTK gene, showing the single CHR site and TTS in Figure 5.*

We appreciate this suggestion and have added a schematic map of the *TTK* promoter fragment to Fig. 5 and the full sequence to Supplemental Figure 5A with the TTS and CHR indicated.

Reviewer 2

1) *It remains somewhat unclear as to how the complex stabilizes nucleosomes in the absence of other factors. The authors propose that MuvB could prevent the unwrapping and movement of the nucleosome. FRET assays with a fluorescently labeled nucleosome might help to clarify this.*

In our model, MuvB stabilizes the nucleosome position because it makes a bipartite interaction with both DNA and the nucleosome directly through the LIN54-CHR interaction and RBAP48 binding, respectively. In this way, MuvBN is able to tether the nucleosome nearby the LIN54-bound CHR. This model is supported by data in Figs. 4, 5 and 6 and our previous study of LIN54-CHR binding (PMID:27465258). We show this model in Fig. 7C and have made effort in the discussion of the revised manuscript to describe this idea more clearly. We have added the following sentences:

We propose that MuvB stabilizes nucleosome position by making a bipartite interaction with the nucleosome and DNA. LIN54 binds the CHR sequence while MuvBN tethers the nucleosome near the CHR through a direct association with the histone tails or additional nucleosome contacts.

We have considered FRET assays, but we believe a bulk FRET assay would most likely generate population data that merely corroborates the main result of our EM bubble assay (Fig. 5), i.e. the presence of MuvB increases the occupancy of nucleosomes. We agree that single molecule FRET may be able to provide information about the effect of MuvB on nucleosome dynamics, and it will be interesting to explore how changes to these dynamics affect chromatin-directed processes such as remodeling. However, developing such an assay takes extensive time and effort, and considering a number of other interesting questions could be asked, we hope the reviewer will agree with us that these experiments could follow this current work in a separate study.

2) *Data shown in Fig. 4 were collected on the MuvB complex, which may complicate analysis of fluorescence curves, particularly in case if there are multiple binding sites and therefore multiple contacts, even weak, involved. It will be clearer and easier to understand contribution of each subunit to the binding through measuring binding affinities of isolated subunits and then comparing with the affinity of the entire complex.*

We are satisfied that the fluorescence polarization studies in Fig. 4 and Supplementary Fig. 4 demonstrate that MuvB directly associates with the H3 tail and nucleosomes and that MuvBN is sufficient for these interactions based on our similar measured affinities. We agree that fluorescence curves may be complicated by multiple interactions and that in fitting these curves we have assumed a single binding affinity. We have added the following sentence to the appropriate figure captions stating this assumption:

Data are fit assuming a single binding constant.

Considering we have not been able to produce and purify LIN9 and LIN37 when expressed individually, it is only possible at this time for us to assay binding of isolated RBAP48. We have added RBAP48 binding data and comparisons to published data as described in more detail in our response to point 3 below.

3) *Related to comment 2, if the association of the complex with the nucleosome is driven primarily by binding of RBAP48, it will be interesting to compare affinities of RBAP48 (published and reported here) for different ligands.*

Unfortunately, we have been unable to perform FP assays with RBAP48, as we observe strong nonspecific binding effects, especially when assaying interactions with nucleosomes. However, we have successfully measured binding of RBAP48 to the histone peptides using isothermal titration calorimetry (ITC) and found that H3 affinity is similar to the affinity of the MuvBN-H3 complex and to affinities previously reported for the RBAP48-H3 complex (reference 47). As expected from our structural data, RBAP48 but not MuvBN binds H4. ITC data have been added to Supplementary Fig. 4B, and the following sentence has been added to the main text:

Isothermal titration calorimetry measurements also demonstrate binding of MuvBN to H3 but not H4 tails and suggest that H3 binding is mediated through RBAP48 as previously described (Supplementary Fig. 4B)⁴⁷⁻⁴⁸.

4) *A minor point- the H3K9me3 is not a mark of transcriptional repression.*

We appreciate the clarification and have changed the description of the mark as follows:

To probe whether post-translational modifications on H3 tails influence MuvB binding, we tested two H3 marks that are associated with active transcription (H3K4me) or transcriptionally silent heterochromatin (H3K9me3).

Reviewer 3

1) *In the introduction it is noted that MuvB is conserved through animals but recent papers suggest that is more widely conserved albeit poorly characterized in protists (ciliates) as well (PMID: 34086947) which is probably worth noting.*

We have noted that components of the complex are conserved in animals and ciliates in the introduction and added a reference for this recent paper. The sentence in the introduction now reads as follows:

The MuvB complex, components of which are evolutionarily conserved throughout animals and ciliates, plays a key role in development and differentiation and is considered a master regulator of cell-cycle dependent gene expression programs^{10,11,20-25}.

2) *In co-IP experiments of Fig1 why only express a small portion of Lin54?*

In our MuvB assembly experiments with recombinant protein (Fig. 1C), we necessarily use the LIN54 fragment because in our hands the full-length construct expressed in Sf9 cells is unstable and prone to degradation. As we noted in the results, this fragment contains the most conserved and structured sequences in the protein. We also emphasize here that in the experiments monitoring MuvB component interactions in HCT116 cells (Fig. 1D), we are detecting full-length LIN54 protein, and those results are consistent with our assembly model.

3) *Fig1C a little difficult to follow may help if lanes were numbered and then referred to by lane number in the text.*

We appreciate the suggestion and have added lane numbers and refer to them in the description of the figure in the main text.

4) Should the co-IP of MuvBN subunits (Supp Fig1) be moved to main text since this is the complex crystallized ?

We feel this data supports the structure in showing that the subcomplex can be assembled and purified to homogeneity, but the experiment is not essential to the logic of the main text. We prefer to leave as supplementary.

5) There is a small band reacting with the flag antibody in the IP lane Fig1D Lin9 WT that I do not think is accounted for although I could be mistaken

We assume this band arises from some nonspecific cross-reactivity with anti-Flag antibody (faint band in other lanes) and some bleeding from the very intense Flag-GFP signal in the adjacent lane. We have now noted the nonspecific bands in this Fig. 1D panel using asterisks.

6) RBAP48 is somewhat enigmatic with respect to function in diverse chromatin complexes - How do the histone H3 and H4 binding surfaces of RBAP48 compare when present in PRC2 to that of muvBN i.e. H4 occluded and H3 available – if known, how about for CAF-1 ?

We added the following sentence to the results section addressing the reviewer's question about PRC2.

With respect to histone binding, the H4 binding site in RBAP48 is occluded and the H3 site is more accessible in both the PRC2 and MuvBN complexes.

7) Does the mutTTK site prevent or change muvB from binding in gel shift assay? i.e. this could correlate with the decreased nucleosomes on the promoter

We added a fluorescence polarization experiment in which we tested MuvB binding to a fragment of the TTK promoter (included as Supplemental Fig. 5C). We found decreased affinity when the CHR site in the fluorescent DNA probe was mutated to the mutTTK sequence used in the EM bubble experiment. As suggested by the reviewer, this result correlates with the decreased nucleosomes observed on the mutTTK promoter (Fig. 5C). We revised the sentence describing the mutTTK experiment in the results section as follows:

When we mutated the CHR site in the TTK promoter, we observed a significant decrease in the average nucleosomes per molecule (Fig. 5C), which is consistent with weaker MuvB binding to the mutated CHR site in the DNA (Supplemental Fig. 5C).

8) Can the authors comment on the difference in strategies of using WT Lin9 MNase ChIP compared to that of the Lin9 4x mutant that does not bind Lin37 and RBAP48 with performing Lin9 MNase ChIP +/- RNAi knockdown of RBAP48

We attempted to knock-out RBAP48 in HCT116 cells, but we recovered no viable clones, suggesting it is essential. In addition, as noted by the reviewer, RBAP48 is present in several chromatin interacting complexes. For these reasons, we worried that interpreting an RNAi knockdown experiment would be problematic, and alternatively, we performed the experiment with a mutant LIN9 that does not bind RBAP48.

9) Figure 7A, bottom panel left; since authors are measuring the difference in overall binding density around TSS (+1 nucleosome), I would suggest that authors perform a statistical test (for example, Mann-Whitney U tests) and calculate p-value (optional). I leave that to authors whether to calculate the significance (p-value) or not in this case, however.

We performed a student's t-test on the aggregate data used in Fig. 7A and the other replicates (Supplementary Fig. 7B). We calculated p values comparing the three replicates performed in both wild-type and LIN37^{-/-} cells. The results of the statistical analysis, shown now in Supplementary Fig. 7B, suggest there are significant differences (p<0.05) in regions 100-200 bp from the TSS on either side. We added this following sentence to the results section:

We observe a significant (p<0.05) difference in nucleosome occupancy comparing the wild-type and LIN37 knockout data sets in regions 100-200 bp both downstream and upstream of the TSS in DREAM genes (Supplementary Fig. 7B).

In addition, we calculated the ratio of integrated peak intensity 200 bp upstream and downstream of the TSS in the sequencing read data for genes that were significantly enriched in the experiments performed in both WT and LIN37^{-/-} HCT116 cells. We found that there is a significantly greater relative intensity in WT cells compared to LIN37^{-/-} cells, i.e. there is more bias to nucleosome protection immediately downstream of the TSS in WT cells, which is consistent with our conclusion that nucleosomes are better positioned around the TSS in WT cells. This analysis, which also includes a statistical analysis, is now shown in Supplementary Fig. 7C.

10) *Ciliates lack Lin37 as indicated in a recent report (PMID: 34086947, see above). Authors might want to mention this. Discussion section will benefit from this as it will highlight the Lin37 role in multicellular eukaryotes.*

As noted above, we added a citation to this paper and mention the presence of MuvB components in ciliates in the introduction. With the intention of keeping our discussion short and focused on the implications of nucleosome binding for MuvB function, we chose not to further expand on potential functions of LIN37 and differences between MuvB in mammals and lower organisms.

REVIEWERS' COMMENTS

Reviewer #1 (Remarks to the Author):

The MuvB complex contains at least 5 proteins and contributes to specific regulation of several hundred cell cycle dependent genes. In the revised manuscript, Rubin and colleagues provide new structural information on the MuvB complex and specifically functions to protect the +1 nucleosome positioned near the TSS of these genes. They propose that the ability of the MuvB complex to enhance the interaction of nucleosomes near the TSS provides a new model for transcription factor repression.

The changes in response to the critiques are thoughtful, appropriate, and relevant. The authors have clarified several points related to the LIN9-4x mutant, PAF-binding to MuvB, and the TTK promoter that have improved explanation of their data

Reviewer #2 (Remarks to the Author):

The authors have adequately addressed my previous comments.

Reviewer #3 (Remarks to the Author):

Hi I am satisfied with the revised version of the manuscript